# A Universally Optimal Multistage Accelerated Stochastic Gradient Method

**Necdet Serhat Aybat**[*]
Pennsylvania State University
University Park, PA, USA
nsa10@psu.edu

**Alireza Fallah**[*]
Massachusetts Institute of Technology
Cambridge, MA, USA
afallah@mit.edu

**Mert Gürbüzbalaban**[*]
Rutgers University
Piscataway, NJ, USA
mg1366@rutgers.edu

**Asuman Ozdaglar**[*]
Massachusetts Institute of Technology
Cambridge, MA, USA
asuman@mit.edu

## Abstract

We study the problem of minimizing a strongly convex, smooth function when we have noisy estimates of its gradient. We propose a novel multistage accelerated algorithm that is universally optimal in the sense that it achieves the optimal rate both in the deterministic and stochastic case and operates without knowledge of noise characteristics. The algorithm consists of stages that use a stochastic version of Nesterov's method with a specific restart and parameters selected to achieve the fastest reduction in the bias-variance terms in the convergence rate bounds.

## 1 Introduction

First order optimization methods play a key role in solving large scale machine learning problems due to their low iteration complexity and scalability with large data sets. In several cases, these methods operate with noisy first order information either because the gradient is estimated from draws or subset of components of the underlying objective function [3, 8, 13, 16, 17, 21, 36, 9, 11] or noise is injected intentionally due to privacy or algorithmic considerations [4, 25, 30, 14, 15]. A fundamental question in this setting is to design fast algorithms with *optimal convergence rate*, matching the lower bounds on the oracle complexity in terms of target accuracy and other important parameters both for the deterministic and stochastic case (i.e., with or without gradient errors).

In this paper, we design an *optimal* first order method to solve the problem

$$f^* \triangleq \min_{x \in \mathbb{R}^d} f(x) \quad \text{such that } f \in S_{\mu, L}(\mathbb{R}^d), \tag{1}$$

where, for scalars $0 < \mu \leq L$, $S_{\mu, L}(\mathbb{R}^d)$ is the set of continuously differentiable functions $f : \mathbb{R}^d \to \mathbb{R}$ that are strongly convex with modulus $\mu$ and have Lipschitz-continuous gradients with constant $L$, which imply that for every $x, y \in \mathbb{R}^d$, $f$ satisfies (see e.g. [27])

$$\frac{\mu}{2}\|x - y\|^2 \leq f(x) - f(y) - \nabla f(y)^\top (x - y) \leq \frac{L}{2}\|x - y\|^2. \tag{2}$$

For $f \in S_{\mu, L}(\mathbb{R}^d)$, the ratio $\kappa \triangleq \frac{L}{\mu}$ is called the *condition number* of $f$. Throughout the paper, we denote the solution of problem (1) by $f^*$ which is achieved at the *unique* optimal point $x^*$.

---

[*]The authors are in alphabetical order.

We assume that the gradient information is available through a stochastic oracle, which at each iteration $n$, given the current iterate $x_n \in \mathbb{R}^d$, provides the noisy gradient $\tilde{\nabla} f(x_n, w_n)$ where $\{w_n\}_n$ is a sequence of independent random variables such that for all $n \geq 0$,

$$\mathbb{E}[\tilde{\nabla} f(x_n, w_n)|x_n] = \nabla f(x_n), \quad \mathbb{E}\left[\|\tilde{\nabla} f(x_n, w_n) - \nabla f(x_n)\|^2 \Big| x_n\right] \leq \sigma^2. \quad (3)$$

This oracle model is commonly considered in the literature (see e.g. [16, 17, 6]). In Appendix K, we show how our analysis can be extended to the following more general noise setting, same as the one studied in [3], where the variance of the noise is allowed to grow linearly with the squared distance to the optimal solution:

$$\mathbb{E}[\tilde{\nabla} f(x_n, w_n)|x_n] = \nabla f(x_n), \quad \mathbb{E}\left[\|\tilde{\nabla} f(x_n, w_n) - \nabla f(x_n)\|^2 \Big| x_n\right] \leq \sigma^2 + \eta^2\|x_n - x^*\|^2. \quad (4)$$

for some constant $\eta \geq 0$.

Under noise setting in (3), the performance of many algorithms is characterized by the expected error of the iterates (in terms of the suboptimality in function values) which admits a bound as a sum of two terms: a *bias term* that shows the decay of initialization error $f(x_0) - f^*$ and is independent of the noise parameter $\sigma^2$, and a *variance term* that depends on $\sigma^2$ and is independent of the initial point $x_0$. A lower bound on the bias term follows from the seminal work of Nemirovsky and Yudin [26], which showed that without noise ($\sigma = 0$) and after $n$ iterations, $\mathbb{E}\left[f(x_n)\right] - f^*$ cannot be smaller than[2]

$$L\|x_0 - x^*\|_2^2 \exp(-\mathcal{O}(1)\frac{n}{\sqrt{\kappa}}). \quad (5)$$

With noise, Raginsky and Rakhlin [31] provided the following (much larger) lower bound[3] on function suboptimality which also provides a lower bound on the variance term:

$$\Omega\left(\frac{\sigma^2}{\mu n}\right) \quad \text{for } n \text{ sufficiently large.} \quad (6)$$

Several algorithms have been proposed in the recent literature attempting to achieve these lower bounds.[4] Xiao [38] obtains $\mathcal{O}(\log(n)/n)$ performance guarantees in expected suboptimality for an accelerated version of the dual averaging method. Dieuleveut et al. [12] consider quadratic objective function and develop an algorithm with averaging to achieve the error bound $\mathcal{O}(\frac{\sigma^2}{n} + \frac{\|x_0 - x^*\|^2}{n^2})$. Hu et al. [20] consider general strongly convex and smooth functions and achieve an error bound with similar dependence under the assumption of bounded noise. Ghadimi and Lan [16] and Chen et al. [7] extend this result to the noise model in (3) by introducing the accelerated stochastic approximation algorithm (AC-SA) and optimal regularized dual averaging algorithm (ORDA), respectively. Both AC-SA and ORDA have multistage versions presented in [17] and [7] where authors improve the bias term of their single stage methods to the optimal $\exp(-\mathcal{O}(1)n/\sqrt{\kappa})$ by exploiting knowledge of $\sigma$ and the optimality gap $\Delta$, i.e., an upper bound for $f(x_0) - f^*$, in the operation of the algorithm. Another closely related paper is [8] which proposed $\mu$AGD+ and showed under *additive* noise model that it admits the error bound $\mathcal{O}(\frac{\sigma^2}{n} + \frac{\|x_0 - x^*\|^2}{n^p})$ for any $p \geq 1$ where the constants grow with $p$, and in particular, they achieve the bound $\mathcal{O}(\frac{\sigma^2 \log n}{n} + \frac{\|x_0 - x^*\|^2 \log n}{n^{\log n}})$ for $p = \log n$.

In this paper, we introduce the class of Multistage Accelerated Stochastic Gradient (M-ASG) methods that are universally optimal, achieving the lower bound both in the noiseless deterministic case and the noisy stochastic case up to some constants independent of $\mu$ and $L$. M-ASG proceeds in stages that use a stochastic version of Nesterov's accelerated method [27] with a specific restart and parameterization. Given an arbitrary length and constant stepsize for the first stage together with geometrically growing lengths and shrinking stepsizes for the following stages, we first provide a general convergence rate result for M-ASG (see Theorem 3.4). Given the computational budget $n$, a specific choice for the length of the first stage is shown to achieve the optimal error bound *without requiring knowledge of the noise bound $\sigma^2$ and the initial optimality gap* (See Corollary 3.8).

Table 1: Comparison of algorithms

| Algorithm | Requires | | | Opt. Bias | Opt. Var. |
|---|---|---|---|---|---|
| | $\sigma$ | $\Delta$ | $n$ or $\epsilon$ | | |
| AC-SA | ✗ | ✗ | ✗ | ✗ | ✓ |
| Multi. AC-SA | ✓ | ✓ | ✗ | ✓ | ✓ |
| ORDA | ✗ | ✗ | ✗ | ✗ | ✓ |
| Multi. ORDA | ✓ | ✓ | ✗ | ✓ | ✓ |
| Cohen et al. | ✗ | ✗ | ✗ | ✗ | ✓ |
| **M-ASG** (With parameters in **Corollary 3.7**) | ✗ | ✗ | ✗ | ✗ | ✓ |
| **M-ASG** (With parameters in **Corollary 3.8**) | ✗ | ✗ | ✓[n] | ✓ | ✓ |
| **M-ASG** (With parameters in **Corollary 3.9**) | ✗ | ✓ | ✓[ε] | ✓ | ✓ |

To the best of our knowledge, this is the first algorithm that achieves such a lower bound under such informational assumptions. In Table 1, we provide a comparison of our algorithm with other algorithms in terms of required assumptions and optimality of their results in both bias and variance terms. In particular, we consider ACSA [16], Multistage AC-SA [17], ORDA and Multistage ORDA [7], and the algorithm proposed in [8].

Our paper builds on an analysis of Nesterov's accelerated stochastic method with a specific momentum parameter presented in Section 2 which may be of independent interest. This analysis follows from a dynamical system representation and study of first order methods which has gained attention in the literature recently [24, 19, 2]. In Section 3, we present the M-ASG algorithm, and characterize its behavior under different assumptions as summarized in Table 1. In particular, we show that it achieves the optimal convergence rate with the given budget of iterations $n$. In Section 4, we show how additional information such as $\sigma$ and $\Delta$ can be leveraged in our framework to improve practical performance. Finally, in Section 5, we provide numerical results on the comparison of our algorithm with some of the other most recent methods in the literature.

**Preliminaries and notation:** Let $I_d$ and $0_d$ represent the $d \times d$ identity and zero matrices. For matrix $A \in \mathbb{R}^{d \times d}$, $\text{Tr}(A)$ and $\det(A)$ denote the trace and determinant of $A$, respectively. Also, for scalars $1 \leq i \leq j \leq d$ and $1 \leq k \leq l \leq d$, we use $A_{[i:j],[k:l]}$ to show the submatrix formed by rows $i$ to $j$ and columns $k$ to $l$. We use the superscript $\top$ to denote the transpose of a vector or a matrix depending on the context. Throughout this paper, all vectors are represented as column vectors. Let $\mathbb{S}_+^m$ denote the set of all symmetric and positive semi-definite $m \times m$ matrices. For two matrices $A \in \mathbb{R}^{m \times n}$ and $B \in \mathbb{R}^{p \times q}$, their Kronecker product is denoted by $A \otimes B$. For scalars $0 < \mu \leq L$, $S_{\mu,L}(\mathbb{R}^d)$ is the set of continuously differentiable functions $f : \mathbb{R}^d \to \mathbb{R}$ that are strongly convex with modulus $\mu$ and have Lipschitz-continuous gradients with constant $L$. All logarithms throughout the paper are in natural basis.

## 2 Modeling Accelerated Gradient method as a dynamical system

In this section we study Nesterov's Accelerated Stochastic Gradient method (ASG) [27] with the stochastic first-order oracle in (3):

$$y_k = (1 + \beta)x_k - \beta x_{k-1}, \quad x_{k+1} = y_k - \alpha \tilde{\nabla} f(y_k, w_k) \tag{7}$$

where $\alpha \in (0, \frac{1}{L}]$ is the stepsize and $\beta = \frac{1 - \sqrt{\alpha\mu}}{1 + \sqrt{\alpha\mu}}$ is the momentum parameter. This choice of momentum parameter has already been studied in the literature, e.g., [28, 37, 33]. In the next lemma, we provide a new motivation for this choice by showing that for quadratic functions and in the noiseless setting, this momentum parameter achieves the *fastest* asymptotic convergence rate for a given fixed stepsize $\alpha \in (0, \frac{1}{L}]$. The proof of this lemma is provided in Appendix A.

**Lemma 2.1.** *Let $f \in S_{\mu,L}(\mathbb{R}^d)$ be a strongly convex quadratic function such that $f(x) = \frac{1}{2}x^\top Q x - p^\top x + r$ where $Q$ is a $d$ by $d$ symmetric positive definite matrix with all its eigenvalues in the interval $[\mu, L]$. Consider the deterministic ASG iterations, i.e., $\sigma = 0$, as shown in (7), with constant stepsize $\alpha \in (0, 1/L]$. Then, the fastest asymptotic convergence rate, i.e. the smallest $\rho \in (0, 1)$ that satisfies the inequality*

$$\|x_k - x^*\|^2 \leq (\rho + \epsilon_k)^{2k} \|x_0 - x^*\|^2, \quad \forall x_0 \in \mathbb{R}^d,$$

for some non-negative sequence $\{\epsilon_k\}_k$ that goes to zero is $\rho = 1 - \sqrt{\alpha\mu}$[5] and it is achieved by $\beta = \frac{1-\sqrt{\alpha\mu}}{1+\sqrt{\alpha\mu}}$. As a consequence, for this choice of $\beta$, there exists $\{\epsilon_k\}$ such that $\lim_{k\to\infty} \epsilon_k = 0$ and

$$f(x_k) - f^* \leq L(1 - \sqrt{\alpha\mu} + \epsilon_k)^{2k}\|x_0 - x^*\|^2.$$

Our analysis builds on the reformulation of a first-order optimization algorithm as a linear dynamical system. Following [24, 19], we write ASG iterations as

$$\xi_{k+1} = A\xi_k + B\tilde{\nabla}f(y_k, w_k), \qquad y_k = C\xi_k, \tag{8}$$

where $\xi_k := \begin{bmatrix} x_k^\top & x_{k-1}^\top \end{bmatrix}^\top \in \mathbb{R}^{2d}$ is the *state* vector and $A$, $B$ and $C$ are matrices with appropriate dimensions defined as the Kronecker products $A = \tilde{A} \otimes I_d$, $B = \tilde{B} \otimes I_d$ and $C = \tilde{C} \otimes I_d$ with

$$\tilde{A} = \begin{bmatrix} 1 + \beta & -\beta \\ 1 & 0 \end{bmatrix}, \quad \tilde{B} = \begin{bmatrix} -\alpha \\ 0 \end{bmatrix}, \quad \tilde{C} = \begin{bmatrix} 1 + \beta & -\beta \end{bmatrix}. \tag{9}$$

We can also relate the state $\xi_k$ to the iterate $x_k$ in a linear fashion through the identity $x_k = T\xi_k$, $\quad T \triangleq \begin{bmatrix} I_d & 0_d \end{bmatrix}$. We study the evolution of the ASG method through the following *Lyapunov function* which also arises in the study of deterministic accelerated gradient methods:

$$V_P(\xi) = (\xi - \xi^*)^\top P(\xi - \xi^*) + f(T\xi) - f^* \tag{10}$$

where $P$ is a symmetric positive semi-definite matrix. We first state the following lemma which can be derived by adapting the proof of Proposition 4.6 in [2] to our setting with less restrictive noise assumption compared to the additive noise model of [2]. Its proof can be found in Appendix B.

**Lemma 2.2.** *Let* $f \in S_{\mu,L}(\mathbb{R}^d)$ . *Consider the ASG iterations given by* (7). *Assume there exist* $\rho \in (0,1)$ *and* $\tilde{P} \in \mathbb{S}_+^2$, *possibly depending on* $\rho$, *such that*

$$\rho^2 \tilde{X}_1 + (1 - \rho^2)\tilde{X}_2 \succeq \begin{bmatrix} \tilde{A}^\top \tilde{P}\tilde{A} - \rho^2\tilde{P} & \tilde{A}^\top \tilde{P}\tilde{B} \\ \tilde{B}^\top \tilde{P}\tilde{A} & \tilde{B}^\top \tilde{P}\tilde{B} \end{bmatrix} \tag{11}$$

*where*

$$\tilde{X}_1 = \frac{1}{2}\begin{bmatrix} \beta^2\mu & -\beta^2\mu & -\beta \\ -\beta^2\mu & \beta^2\mu & \beta \\ -\beta & \beta & \alpha(2 - L\alpha) \end{bmatrix}, \quad \tilde{X}_2 = \frac{1}{2}\begin{bmatrix} (1+\beta)^2\mu & -\beta(1+\beta)\mu & -(1+\beta) \\ -\beta(1+\beta)\mu & \beta^2\mu & \beta \\ -(1+\beta) & \beta & \alpha(2 - L\alpha) \end{bmatrix}.$$

*Let* $P = \tilde{P} \otimes I_d$. *Then, for every* $k \geq 0$,

$$\mathbb{E}\left[V_P(\xi_{k+1})\right] \leq \rho^2 \mathbb{E}\left[V_P(\xi_k)\right] + \sigma^2\alpha^2(\tilde{P}_{1,1} + \frac{L}{2}). \tag{12}$$

We use this lemma and derive the following theorem which characterize the behavior of ASG method for when $\alpha \in (0, 1/L]$ and $\beta = \frac{1-\sqrt{\alpha\mu}}{1+\sqrt{\alpha\mu}}$ (see the proof in Appendix C).

**Theorem 2.3.** *Let* $f \in S_{\mu,L}(\mathbb{R}^d)$ . *Consider the ASG iterations given in* (7) *with* $\alpha \in (0, \frac{1}{L}]$ *and* $\beta = \frac{1-\sqrt{\alpha\mu}}{1+\sqrt{\alpha\mu}}$. *Then,*

$$\mathbb{E}\left[V_{P_\alpha}(\xi_{k+1})\right] \leq (1 - \sqrt{\alpha\mu})\mathbb{E}\left[V_{P_\alpha}(\xi_k)\right] + \frac{\sigma^2\alpha}{2}(1 + \alpha L) \tag{13}$$

*for every* $k \geq 0$, *where* $P_\alpha = \tilde{P}_\alpha \otimes I_d$ *with* $\tilde{P}_\alpha = \begin{bmatrix} \sqrt{\frac{1}{2\alpha}} \\ \sqrt{\frac{\mu}{2}} - \sqrt{\frac{1}{2\alpha}} \end{bmatrix}\begin{bmatrix} \sqrt{\frac{1}{2\alpha}} & \sqrt{\frac{\mu}{2}} - \sqrt{\frac{1}{2\alpha}} \end{bmatrix}$.

This result relies on the special structure of $P_\alpha$ which will also be key for our analysis in Section 3.

## 3 A class of multistage ASG algorithms

In this section, we introduce a class of *multistage* ASG algorithms, represented in Algorithm 1 which we denote by M-ASG. The main idea is to run ASG with properly chosen parameters $(\alpha_k, \beta_k)$ at

---
**Algorithm 1:** Multistage Accelerated Stochastic Gradient Algorithm (M-ASG)
---
1 Set $n_0 = -1$;
2 **for** $k = 1;\ k \leq K;\ k = k + 1$ **do**
3 $\quad$ Set $x_0^k = x_1^k = x_{n_{k-1}+1}^{k-1}$;
4 $\quad$ **for** $m = 1;\ m \leq n_k;\ m = m + 1$ **do**
5 $\quad\quad$ Set $\beta_k = \frac{1 - \sqrt{\mu\alpha_k}}{1 + \sqrt{\mu\alpha_k}}$;
6 $\quad\quad$ Set $y_m^k = (1 + \beta_k)x_m^k - \beta_k x_{m-1}^k$;
7 $\quad\quad$ Set $x_{m+1}^k = y_m^k - \alpha_k \hat{\nabla} f(y_m^k, w_m^k)$
8 $\quad$ **end**
9 **end**
---

each stage $k \in 1, \dots, K$ for $K \geq 2$ stages. In addition, each new stage is dependent on the previous stage as the first two initial iterates of the new stage are set to the last iterate of the previous stage.

To analyze Algorithm 1, we first characterize the evolution of iterates in one specific stage through the Lyapunov function in (10). The details of the proof is provided in Appendix D.

**Theorem 3.1.** *Let $f \in S_{\mu,L}(\mathbb{R}^d)$. Consider running the ASG method given in (7) for $n$ iterations with $\alpha = \frac{c^2}{L}$ and $\beta = \frac{1-\sqrt{\alpha\mu}}{1+\sqrt{\alpha\mu}}$ for some $0 < c \leq 1$. Then, for $P_\alpha$ given in Theorem 2.3,*

$$\mathbb{E}\left[V_{P_\alpha}(\xi_{n+1})\right] \leq \exp(-n\frac{c}{\sqrt{\kappa}})\mathbb{E}\left[V_{P_\alpha}(\xi_1)\right] + \frac{\sigma^2\sqrt{\kappa}c}{L}. \tag{14}$$

Given a computational budget of $n$ iterations, we use this result to choose a stepsize that help us achieve an approximately optimal decay in the *variance term* which yields the following corollary for M-ASG algorithm with $K = 1$ stage, and its proof can be found in Appendix E.

**Corollary 3.2.** *Let $f \in S_{\mu,L}(\mathbb{R}^d)$. Consider running M-ASG, i.e., Algorithm 1, for only one stage with $n_1 = n$ iterations and stepsize $\alpha_1 = \left(\frac{p\sqrt{\kappa}\log n}{n}\right)^2 \frac{1}{L}$ for some scalar $p \geq 1$. Then,*

$$\mathbb{E}\left[f(x_{n+1}^1)\right] - f^* \leq \frac{2}{n^p}(f(x_0^0) - f^*) + \frac{p\sigma^2\log n}{n\mu} \tag{15}$$

*provided that $n \geq p\sqrt{\kappa}\max\{2\log(p\sqrt{\kappa}), e\}$.*

For subsequent analysis, given $K \geq 1$, for all $1 \leq k \leq K$, we define the state vector $\xi_i^k = \left[x_i^{k\top}, x_{i-1}^{k\top}\right]^\top$ for $1 \leq i \leq n_k + 1$ –recall that $x_0^k = x_1^k = x_{n_{k-1}+1}^{k-1}$, where $K$ is the number of stages. We analyze the performance of each stage with respect to a stage-dependent Lyapunov function $V_{P_{\alpha_k}}$. The following lemma relates the performance bounds with respect to consecutive choice of Lyapunov functions, building on our specific restarting mechanism (The proof can be found in Appendix F).

**Lemma 3.3.** *Let $f \in S_{\mu,L}(\mathbb{R}^d)$. Consider M-ASG, i.e., Algorithm 1. Then, for every $1 \leq k \leq K-1$,*

$$\mathbb{E}\left[V_{P_{\alpha_{k+1}}}(\xi_1^{k+1})\right] \leq 2\mathbb{E}\left[V_{P_{\alpha_k}}(\xi_{n_k+1}^k)\right]. \tag{16}$$

Now, we are ready to state and prove the main result of the paper (see proof in Appendix G):

**Theorem 3.4.** *Let $f \in S_{\mu,L}(\mathbb{R}^d)$. Consider running M-ASG, i.e., Algorithm 1, with some $n_1 \geq 1$ and $\alpha_1 = \frac{1}{L}$ and fixing $n_k = 2^k\lceil\sqrt{\kappa}\log(2^{p+2})\rceil$ and $\alpha_k = \frac{1}{2^{2k}L}$ for any $k \geq 2$ and $p \geq 1$. The last iterate of each stage, i.e., $x_{n_k+1}^k$, satisfies the following bound for all $k \geq 1$:*

$$\mathbb{E}\left[f(x_{n_k+1}^k)\right] - f^* \leq \frac{2}{2^{(p+1)(k-1)}}\left(\exp(-n_1/\sqrt{\kappa})(f(x_0^0) - f^*)\right) + \frac{\sigma^2\sqrt{\kappa}}{L2^{k-1}}. \tag{17}$$

We next define $N_K(p, n_1)$ as the number of iterations needed to run M-ASG for $K \geq 1$ stages, i.e., $N_K(p, n_1) \triangleq \sum_{k=1}^{K} n_k$. Note for $K \geq 2$ and with parameters given in Theorem 3.4,

$$N_K(p, n_1) = n_1 + (2^{K+1} - 4)\lceil \sqrt{\kappa} \log(2^{p+2}) \rceil. \tag{18}$$

We define $\{x_n\}_{n \in \mathbb{Z}_+}$ sequence such that $x_n$ is the iterate generated by M-ASG algorithm at the end of $n$ gradient steps for $n \geq 0$, i.e., $x_0 = x_0^0$, $x_n = x_{n+1}^1$ for $1 \leq n \leq n_1$, and for $n > n_1$ we set $x_n = x_m^k$ where $k = \lceil \log_2 \left( \frac{n - n_1}{\lceil \sqrt{\kappa} \log(2^{p+2}) \rceil} + 4 \right) - 1 \rceil$ and $m = n - N_{k-1}(p, n_1)$.

**Remark 3.5.** *In the absence of noise, i.e., $\sigma = 0$, the result of Theorem 3.4 recovers the linear convergence rate of deterministic gradient methods as its special case. Indeed, running M-ASG only for one stage with $n$ iterations, i.e., $K = 1$ and $n_1 = n$ guarantees that $\mathbb{E}[f(x_n)] - f^* \leq 2\exp(-n/\sqrt{\kappa})(f(x_0^0) - f^*)$ for all $n \geq 1$.*

The next theorem remarks the behavior of M-ASG after running it for $n$ iterations with the parameters in the preceding theorem, and its proof is provided in Appendix H.

**Theorem 3.6.** *Let $f \in S_{\mu,L}(\mathbb{R}^d)$. Consider running Algorithm 1 for $n$ iterations and with parameters given in Theorem 3.4 and $n_1 < n$. Then the error is bounded by*

$$\mathbb{E}[f(x_n)] - f^*$$
$$\leq \mathcal{O}(1) \left( \frac{(8(p+1)\sqrt{\kappa}\log(2))^{p+1}}{(n-n_1)^{p+1}} \left( \exp(-n_1/\sqrt{\kappa})(f(x_0^0) - f^*) \right) + \frac{(p+1)\sigma^2}{(n-n_1)\mu} \right). \tag{19}$$

**Corollary 3.7.** *Under the premise of Theorem 3.6, choosing $n_1 = \lceil (p+1)\sqrt{\kappa}\log(12(p+1)\kappa) \rceil$, the suboptimality error of M-ASG after $n \geq 2n_1$ admits*

$$\mathbb{E}[f(x_n)] - f^* \leq \mathcal{O}(1) \left( \frac{1}{n^{p+1}}(f(x_0^0) - f^*) + \frac{(p+1)\sigma^2}{n\mu} \right).$$

Theorem 3.6 immediately yields the result in Corollary 3.7, (suboptimal with respect to dependence on initial optimality gap); see Appendix I for the proof. Similar rate results have also been obtained by AC-SA [16] and ORDA [7] algorithms.

We continue this section by pointing out some important special cases of our result. We first show in the next corollary how our algorithm is universally optimal and capable of achieving the lower bounds (5) and (6) simultaneously. The proof follows from (19) and $n - n_1 \geq \frac{n}{2} \geq \sqrt{\kappa}$.

**Corollary 3.8.** *Under the premise of Theorem 3.6, consider a computational budget of $n \geq 2\sqrt{\kappa}$ iterations. By setting $n_1 = \frac{n}{C}$ for some positive constant $C \geq 2$, we obtain a bound matching the lower bounds in (5) and (6), i.e.,*

$$\mathbb{E}[f(x_n)] - f^* \leq \mathcal{O}(1) \left( \exp(-\frac{n}{C\sqrt{\kappa}})(f(x_0^0) - f^*) + \frac{\sigma^2}{n\mu} \right).$$

We note that achieving the lower bound through the M-ASG algorithm requires the knowledge or estimation of the strong convexity constant $\mu$. In some applications, $\mu$ may not be known a priori. However, for regularized risk minimization problems, the regularization parameter is known and it determines the strong convexity constant. It is also worth noting that, even for the deterministic case, [1] has shown that for a wide class of algorithms including ASG, it is *not possible* to obtain the lower bound (5) without knowing the strong convexity parameter. In addition, in Appendix L, we show how our framework can be extended to obtain nearly optimal results in the merely convex setting; i.e. when $\mu = 0$. Finally, note that the Lipschitz constant $L$ can be estimated from data using standard line search techniques in practice, see [5] and [32, Alg. 2].

The lower bound can also be stated as the minimum number of iterations needed to find an $\epsilon-$solution, i.e, to find $x_\epsilon$ such that $\mathbb{E}[f(x_\epsilon)] - f^* \leq \epsilon$, for any given $\epsilon > 0$. In the following corollary, and with the additional assumption of knowing the bound $\Delta$ on the initial optimality gap $f(x_0^0) - f^*$, we state this version of lower bound. The proof is provided in Appendix J.

**Corollary 3.9.** *Let $f \in S_{\mu,L}(\mathbb{R}^d)$. Given $\Delta \geq f(x_0^0) - f^*$, for any $\epsilon \in (0, \Delta)$, running M-ASG, Algorithm 1, with parameters given in Theorem 3.4, $p = 1$, and $n_1 = \lceil \sqrt{\kappa}\log\left(\frac{4\Delta}{\epsilon}\right) \rceil$, one can compute $\epsilon-$solution within $n_\epsilon$ iterations in total, where*

$$n_\epsilon = \lceil \sqrt{\kappa}\log\left(\frac{4\Delta}{\epsilon}\right) \rceil + \lceil 16(1 + \log(8))\frac{\sigma^2}{\mu\epsilon} \rceil. \tag{20}$$

Recall that we presented a comparison with other state-of-the-art algorithms in Table 1. In particular, this table shows that Multistage AC-SA [17] and Multistage ORDA [7] also achieve the lower bounds provided that noise parameters are known – *note we do not make this extra assumption for M-ASG*. It is also worth noting that the idea of restart, which plays a key role in achieving the lower bounds, has been studied before in the context of deterministic accelerated methods [29, 39]. However, a naive extension of these restart methods to the stochastic setting leads to a two-stage algorithm which switches from constant step-size to diminishing step-size when the variance term dominates the bias term. Nevertheless, implementing this technique requires the knowledge of $\sigma^2$ and optimality gap to tune algorithms for achieving optimal rates in both bias and variance terms. M-ASG, on the other hand, achieves the optimal rates using a specific multistage scheme that does not require the knowledge of the parameter $\sigma^2$. In the supplementary material, we also discuss how M-ASG is related to AC-SA and Multistage AC-SA algorithms proposed in [16, 17].

## 4 M-ASG$^*$: An improved bias-variance trade-off

In section 3, we described a universal algorithm that do not require the knowledge of neither initial suboptimality gap $\Delta$ nor the noise magnitude $\sigma^2$ to operate. However, as we will argue in this section, our framework is flexible in the sense that additional information about the magnitude of $\Delta$ or $\sigma^2$ can be leveraged to improve practical performance. We first note that several algorithms in the literature assume that an upper bound on $\Delta$ is known or can be estimated, as summarized in Table 1. This assumption is reasonable in a variety of applications when there is a natural lower bound on $f$. For example, in supervised learning scenarios such as support vector machines, regression or logistic regression problems, the loss function $f$ has non-negative values [35]. Similarly, the noise level $\sigma^2$ may be known or estimated, e.g., in private risk minimization [4], the noise is added by the user to ensure privacy; therefore, it is a known quantity.

There is a natural well-known trade-off between constant and decaying stepsizes (decaying with the number of iterations $n$) in stochastic gradient algorithms. Since the noise is multiplied with the stepsize, a stepsize that is decaying with the number of iterations $n$ leads to a decay in the variance term; however, this will slow down the decay of the bias term, which is controlled essentially by the behavior of the underlying *deterministic accelerated gradient* algorithm (AG) that will give the best performance with the constant stepsize (note that when $\sigma = 0$, the bias term gives the known performance bounds for the AG algorithm). The main idea behind the M-ASG algorithm (which allows it to achieve the lower bounds) is to exploit this trade-off to decide on the *right time*, $n_1$, to *switch* to decaying stepsizes, i.e., when the bias term is sufficiently small so that the variance term dominates and should be handled with the decaying stepsize. This insight is visible from the results of Theorem 3.4 which gives further insights on the choice of the stepsize at every stage to achieve the lower bounds. Theorem 3.4 shows that if M-ASG is run with a constant stepsize $\alpha_1 = \frac{1}{L}$ in the first stage, then the variance term admits the bound $\frac{\sigma^2 \sqrt{\kappa}}{L}$ which does not decay with the number of iterations $n_1$ in the first stage. However, in later stages, when $n > n_1$, the stepsize $\alpha_k$ is decreased as the number of iterations grows and this results in a decay of the variance term. Overall, the choice of the length of the first stage $n_1$, has a major impact in practice which we will highlight in our numerical experiments.

If an estimate of $\Delta$ or $\sigma^2$ is known, it is desirable to choose $n_1$ as small as possible such that it ensures the bias term becomes smaller than the variance term at the end of the first stage. More specifically, applying Theorem 3.1 for $c = 1$, one can choose $n_1$ to balance the variance $\frac{\sigma^2 \sqrt{\kappa}}{L}$ and the bias $\exp(-n_1 \frac{1}{\sqrt{\kappa}}) \mathbb{E}\left[V_{P_{\alpha_1}}(\xi_1^1)\right]$ terms. The term $\mathbb{E}\left[V_{P_{\alpha_1}}(\xi_1^1)\right]$, as shown in the proof of Lemma 3.3, can be bounded by $\mathbb{E}\left[V_{P_{\alpha_1}}(\xi_1^1)\right] = \mu\|x_0^0 - x^*\|_2^2 + f(x_0^0) - f^* \leq 2(f(x_0^0) - f^*)$. Therefore, by having an estimate of an upper bound for $\Delta$, $n_1$ can be set to be the smallest number such that $2\Delta \exp(-n_1 \frac{1}{\sqrt{\kappa}}) \leq \frac{\sigma^2 \sqrt{\kappa}}{L}$, i.e.,

$$n_1 = \lceil \sqrt{\kappa} \log\left(\frac{2L\Delta}{\sigma^2 \sqrt{\kappa}}\right) \rceil. \tag{21}$$

This result allows one to fine-tune the switching point to start using the decaying stepsizes within our framework as a function of $\sigma^2$ and $\Delta$. In scenarios, when the noise level $\sigma$ is small or the initial gap $\Delta$ is large, $n_1$ is chosen large enough to guarantee a fast decay in the bias term. We would like to emphasize that this modified M-ASG algorithm only requires the knowledge of $\sigma$ and $\Delta$ for selecting

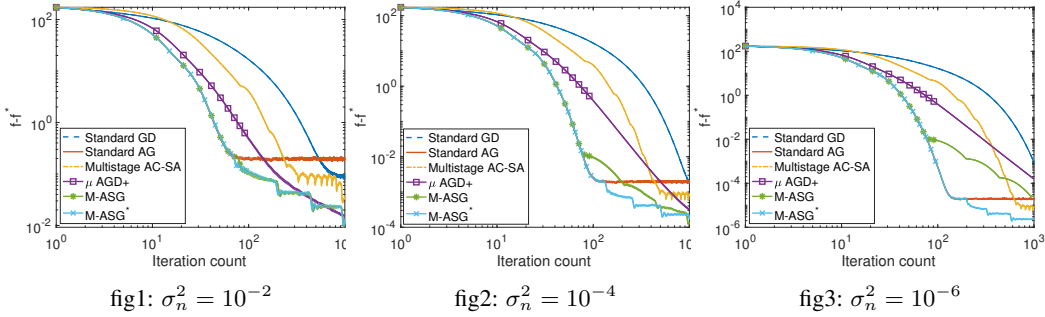

fig1: $\sigma_n^2 = 10^{-2}$        fig2: $\sigma_n^2 = 10^{-4}$        fig3: $\sigma_n^2 = 10^{-6}$

Figure 1: Comparison on a quadratic function for $n = 1000$ iterations with different level of noise.

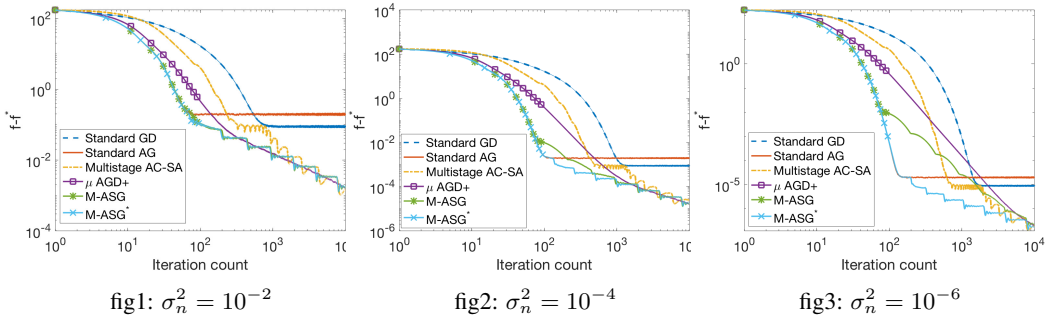

fig1: $\sigma_n^2 = 10^{-2}$        fig2: $\sigma_n^2 = 10^{-4}$        fig3: $\sigma_n^2 = 10^{-6}$

Figure 2: Comparison on a quadratic function for $n = 10000$ iterations with different level of noise.

$n_1$ and the rest of the parameters can be chosen as in Theorem 3.4 which are independent of both $\sigma$ and $\Delta$. Finally, the following theorem provides theoretical guarantees of our framework for this choice of $n_1$. The proof is omitted as it is similar to the proofs of Theorems 3.4 and 3.6.

**Theorem 4.1.** *Let $f \in S_{\mu,L}(\mathbb{R}^d)$. Consider running Algorithm 1 for $n$ iterations and with parameters given in Theorem 3.4, $p = 1$, and $n_1$ set as (21). Then, the expected suboptimality in function values admits the bound $\mathbb{E}\left[f(x_n)\right] - f^* \leq 36(1 + \log(8))\frac{\sigma^2}{(n-n_1)\mu}$ for all $n \geq n_1$.*

## 5    Numerical experiments

In this section, we demonstrate the numerical performance of Algorithm 1 with parameters specified by Corollary 3.7 (M-ASG) and Theorem 4.1 (M-ASG*) and compare with other methods from the literature. In our first experiment, we consider the strongly convex quadratic objective $f(x) = \frac{1}{2}x^\top Q x - bx + \lambda\|x\|^2$ where $Q$ is the Laplacian of a cycle graph[6], $b$ is a random vector and $\lambda = 0.01$ is a regularization parameter. We assume the gradients $\nabla f(x)$ are corrupted by additive noise with a Gaussian distribution $\mathcal{N}(0, \sigma_n^2)$ where $\sigma_n^2 \in \{10^{-6}, 10^{-4}, 10^{-2}\}$. We note that this example has been previously considered in the literature as a problem instance where *Standard ASG* (ASG iterations with standard choice of parameters $\alpha = \frac{1}{L}$ and $\beta = \frac{\sqrt{\kappa}-1}{\sqrt{\kappa}+1}$) perform badly compared to *Standard GD* (Gradient Descent with standard choice of the stepsize $\alpha = 1/L$) [18]. In Figures 1 and 2, we compare M-ASG and M-ASG* with Standard GD, Standard AG, $\mu$AGD+ [8], and Multistage AC-SA [17]. We consider dimension $d = 100$ and initialize all the methods from $x_0^0 = 0$. We run the algorithms Multistage AC-SA, and M-ASG*, having access to the same estimate of $\Delta$. Figures 1-2 show the average performance of all the algorithms along with the $95\%$ confidence interval over 50 sample runs while the total number of iterations $n = 1000$ and $n = 10000$ respectively as the noise level $\sigma^2$ is varied. The simulation results reveal that both M-ASG and M-ASG* have typically a faster decay of the error in the beginning and outperforms the other algorithms in general when

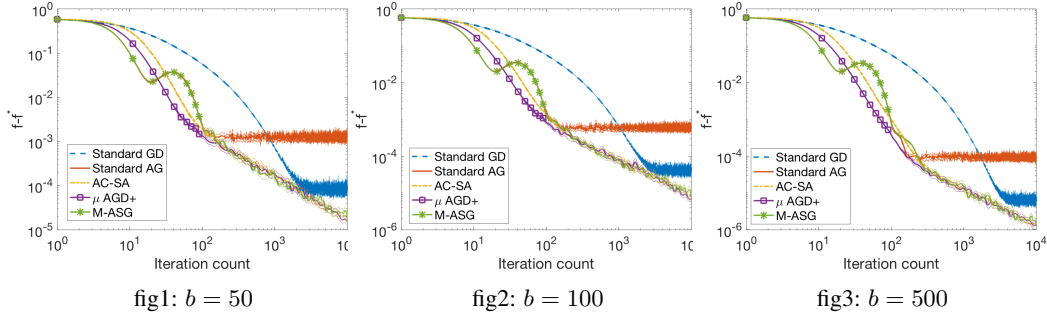

| fig1: $b = 50$ | fig2: $b = 100$ | fig3: $b = 500$ |

Figure 3: Comparison on logistic regression with $n = 10000$ iterations and with different batch sizes.

the number of iterations is small to moderate. In this case, the speed-up obtained by M-ASG and M-ASG* is more prominent if the noise level $\sigma^2$ is smaller. However, as the number of iterations grows, the performance of the algorithms become similar as the variance term dominates. In addition, we would like to highlight that when the noise is small, using $n_1$ as suggested in (21), M-ASG* runs stage one longer than M-ASG; hence, enjoys the linear rate of decay for more iterations before the variance term becomes the dominant term.

For the second set of experiments, we consider a regularized logistic regression problem for binary classification. In particular, we read 10000 images from the M-NIST [23] data-set, and our goal is to distinguish the image of digit zero from that of digit eight.[7] The number of samples is $N = 1945$, and the size of each image is 20 by 20 after removing the margins (hence $d = 400$ after vectorizing the images). At each iteration, we randomly choose a batch size $b$ of images to compute an estimate of the gradient.[8] We choose the regularization parameter equal to $\frac{1}{\sqrt{N}}$ following the standard practice (see e.g. [34]). In Figure 3, we compare M-ASG with Standard GD, Standard AG, $\mu$AGD+ [8], and AC-SA [17] for $b \in \{50, 100, 500\}$. The batch size controls the noise level, with larger batches leading to smaller $\sigma$. We run each of these algorithms for 50 times, and plot their average performance and 95% confidence intervals. It can be seen that M-ASG usually start faster, and achieves the asymptotic rate of other algorithms for all different batch sizes.

# 6  Conclusion

In this work, we consider strongly convex smooth optimization problems where we have access to noisy estimates of the gradients. We proposed a multistage method that adapts the choice of the parameters of the Nesterov's accelerated gradient at each stage to achieve the optimal rate. Our method is universal in the sense that it does not require the knowledge of the noise characteristics to operate and can achieve the optimal rate both in the deterministic and stochastic settings. We provided numerical experiments that compare our method with existing approaches in the literature, illustrating that our method performs well in practice.

## Acknowledgements

The work of Necdet Serhat Aybat is partially supported by NSF Grant CMMI-1635106. Alireza Fallah is partially supported by Siebel Scholarship. Mert Gürbüzbalaban acknowledges support from the grants NSF DMS-1723085 and NSF CCF-1814888.

## Footnotes

[2]This lower bound is shown with the additional assumption $n \leq d$

[3]The authors show this result for $\mu = 1$. Nonetheless, it can be generalized to any $\mu > 0$ by scaling the problem parameters properly.

[4]Here we review their error bounds after $n$ iterations highlighting dependence on $\sigma^2$, $n$, and initial point $x_0$, suppressing $\mu$ and $L$ dependence.

[5]Note that although this rate is asymptotic, its smaller than the non-asymptotic rate that we provide for general strongly convex functions in Theorem 2.3, as there $\rho = \sqrt{1 - \sqrt{\alpha\mu}}$.

[6]All diagonal entries of $Q$ are 2, $Q_{i,j} = -1$ if $|i - j| \equiv 1 \pmod{d}$, and the remaining entries are zero.

[7]We provide an experiment with synthetic data for logistic loss in Appendix N.

[8]This is an unbiased estimate of the gradient with finite but unknown variance, and therefore we do not use M-ASG* or other algorithms that need the knowledge of variance.

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
