[Supplementary Material]

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

# A Proof of Lemma 2.1

Let us denote the asymptotic convergence rate of the ASG method as a function of $\alpha$ and $\beta$ by $\rho(\alpha, \beta)$. It is well-known that $\rho(\alpha, \beta)$ has the following characterization (see e.g. [24], [29]):

$$\rho(\alpha, \beta) = \max\{\rho_\mu(\alpha, \beta), \rho_L(\alpha, \beta)\}, \tag{22}$$

where $\lambda \in \{\mu, L\}$ and $\rho_\lambda$ is defined as:

$$\rho_\lambda(\alpha, \beta) = \begin{cases} \frac{1}{2}|(1+\beta)(1-\alpha\lambda)| + \frac{1}{2}\sqrt{\Delta_\lambda} & \text{if } \Delta_\lambda \geq 0, \\ \sqrt{\beta(1-\alpha\lambda)} & \text{otherwise,} \end{cases} \tag{23}$$

with $\Delta_\lambda = (1+\beta)^2(1-\alpha\lambda)^2 - 4\beta(1-\alpha\lambda)$. Note that, since $\alpha \leq \frac{1}{L}$, we have $1 - \alpha\lambda \geq 0$ for $\lambda \in \{\mu, L\}$; therefore, $\Delta_\lambda \geq 0$ if and only if $(1+\beta)^2(1-\alpha\lambda) \geq 4\beta$, which is equivalent to $\frac{1-\sqrt{\alpha\lambda}}{1+\sqrt{\alpha\lambda}} \geq \beta$.

Using the fact that $\mu \leq L$ and $\frac{1-\sqrt{\alpha\lambda}}{1+\sqrt{\alpha\lambda}}$ is decreasing in $\lambda > 0$, we obtain $\frac{1-\sqrt{\alpha\mu}}{1+\sqrt{\alpha\mu}} \geq \frac{1-\sqrt{\alpha L}}{1+\sqrt{\alpha L}}$; hence, for $\beta > \frac{1-\sqrt{\alpha\mu}}{1+\sqrt{\alpha\mu}}$, we have both $\Delta_\mu < 0$ and $\Delta_L < 0$. As a consequence, (22) implies that for $\beta > \frac{1-\sqrt{\alpha\mu}}{1+\sqrt{\alpha\mu}}$, we have

$$\rho(\alpha, \beta) = \max\{\sqrt{\beta(1-\alpha\mu)}, \sqrt{\beta(1-\alpha L)}\} = \sqrt{\beta(1-\alpha\mu)}. \tag{24}$$

Moreover, for $\beta = \frac{1-\sqrt{\alpha\mu}}{1+\sqrt{\alpha\mu}}$, the two branches in (23) take the same value for $\lambda = \mu$ and $\alpha \in (0, 1/L]$; therefore, when $\beta$ is set to this critical value, we also get $\rho(\alpha, \beta) = \sqrt{\beta(1-\alpha\mu)}$ for $\alpha \in (0, 1/L]$. Note (24) is an increasing function of $\beta$ for any $\alpha \in (0, 1/L]$; thus, given $\alpha \in (0, 1/L]$, the smallest rate possible is equal to $\inf\{\rho(\alpha, \beta) : \beta \geq \frac{1-\sqrt{\alpha\mu}}{1+\sqrt{\alpha\mu}}\} = 1 - \sqrt{\alpha\mu}$, which is the rate given in the statement of the lemma and it is achieved by $\beta = \frac{1-\sqrt{\alpha\mu}}{1+\sqrt{\alpha\mu}}$.

Now, we consider the case $\beta \leq \frac{1-\sqrt{\alpha\mu}}{1+\sqrt{\alpha\mu}}$. From (22), if $\rho_\mu(\alpha, \beta) \geq 1 - \sqrt{\alpha\mu}$, then we also have $\rho(\alpha, \beta) \geq 1 - \sqrt{\alpha\mu}$. Thus, showing $\rho_\mu(\alpha, \beta) \geq 1 - \sqrt{\alpha\mu}$ suffices us to claim that for any $\alpha \in (0, 1/L]$, the best possible rate is $1 - \sqrt{\alpha\mu}$ and this can be achieved by setting $\beta = \frac{1-\sqrt{\alpha\mu}}{1+\sqrt{\alpha\mu}}$. Indeed, as we discussed above, for the case $\beta \leq \frac{1-\sqrt{\alpha\mu}}{1+\sqrt{\alpha\mu}}$, we have $\Delta_\mu \geq 0$; thus,

$$\begin{aligned}
\rho_\mu(\alpha, \beta) &= \frac{1}{2}(1+\beta)(1-\alpha\mu) + \frac{1}{2}\sqrt{\Delta_\mu} \\
&= \frac{1-\sqrt{\alpha\mu}}{2}\left((1+\beta)(1+\sqrt{\alpha\mu}) + \sqrt{(1+\beta)^2(1+\sqrt{\alpha\mu})^2 - \frac{4\beta(1+\sqrt{\alpha\mu})}{1-\sqrt{\alpha\mu}}}\right).
\end{aligned}$$

Therefore, to show $\rho_\mu(\alpha, \beta) \geq 1 - \sqrt{\alpha\mu}$, we just need to prove

$$\sqrt{(1+\beta)^2(1+\sqrt{\alpha\mu})^2 - \frac{4\beta(1+\sqrt{\alpha\mu})}{1-\sqrt{\alpha\mu}}} \geq 2 - (1+\beta)(1+\sqrt{\alpha\mu}). \tag{25}$$

Taking the square of both sides of (25), it follows that (25) is equivalent to

$$(1+\beta)(1+\sqrt{\alpha\mu}) \geq 1 + \frac{\beta(1+\sqrt{\alpha\mu})}{1-\sqrt{\alpha\mu}}$$

and this holds when $\beta \leq \frac{1-\sqrt{\alpha\mu}}{1+\sqrt{\alpha\mu}}$. Therefore, for any $\alpha \in (0, 1/L]$, we have $\rho_\mu(\alpha, \beta) \geq 1 - \sqrt{\alpha\mu}$ for $\beta \leq \frac{1-\sqrt{\alpha\mu}}{1+\sqrt{\alpha\mu}}$. which completes the proof.

# B Proof of Lemma 2.2

We first state the following lemma which is an extension of Lemma 4.1 in [2] for ASG.

**Lemma B.1.** *Let $P = \tilde{P} \otimes I_d$ where $\tilde{P} \in \mathbb{S}_+^2$ and consider the function $\mathcal{W}_P(\xi) = (\xi - \xi^*)^\top P(\xi - \xi^*)$. Then we have*

$$\mathbb{E}[\mathcal{W}_P(\xi_{k+1})] \leq \mathbb{E}\left[ \begin{bmatrix} \xi_k - \xi^* \\ \nabla f(y_k) \end{bmatrix}^\top \begin{bmatrix} A^\top PA & A^\top PB \\ B^\top PA & B^\top PB \end{bmatrix} \begin{bmatrix} \xi_k - \xi^* \\ \nabla f(y_k) \end{bmatrix} \right] + \sigma^2 \alpha^2 \tilde{P}_{11}. \qquad (26)$$

*Proof.* Let $\tilde{\xi}_k = \xi_k - \xi^*$ for any $k \geq 0$. Since $\xi^* = A\xi^*$, (8) implies $\tilde{\xi}_{k+1} = A\tilde{\xi}_k + B(\tilde{\nabla} f(y_k, w_k))$ for $k \geq 0$. Note that, for any $k \geq 1$, $\tilde{\xi}_k$ and $y_k$ are deterministic functions of $\xi_0, \{w_i\}_{i=0}^{k-1}$. Using this fact, along with knowing that $w_k$ is independent of $\xi_0, \{w_i\}_{i=0}^{k-1}$, implies that

$$\mathbb{E}[\mathcal{W}_P(\xi_{k+1})] = \mathbb{E}[\tilde{\xi}_{k+1}^\top P \tilde{\xi}_{k+1}]$$

$$= \mathbb{E}\left[ (A\tilde{\xi}_k + B\tilde{\nabla} f(y_k, w_k))^\top P(A\tilde{\xi}_k + B\tilde{\nabla} f(y_k, w_k)) \right]$$

$$= \mathbb{E}\left[ \mathbb{E}\left[ (A\tilde{\xi}_k + B\tilde{\nabla} f(y_k, w_k))^\top P(A\tilde{\xi}_k + B\tilde{\nabla} f(y_k, w_k)) \Big| \xi_0, \{w_i\}_{i=0}^{k-1} \right] \right]$$

$$= \mathbb{E}\left[ (A\tilde{\xi}_k + B\nabla f(y_k))^\top P(A\tilde{\xi}_k + B\nabla f(y_k)) \right]$$

$$\quad + \mathbb{E}\left[ \mathbb{E}\left[ \tilde{\nabla} f(y_k, w_k)^\top B^\top PB \tilde{\nabla} f(y_k, w_k) \Big| y_k \right] - \nabla f(y_k)^\top B^\top PB \nabla f(y_k) \right]$$
$$\tag{27}$$

$$= \mathbb{E}\left[ (A\tilde{\xi}_k + B\nabla f(y_k))^\top P(A\tilde{\xi}_k + B\nabla f(y_k)) \right]$$

$$\quad + \alpha^2 \tilde{P}_{11} \mathbb{E}\left[ \mathbb{E}\left[ \tilde{\nabla} f(y_k, w_k)^\top \tilde{\nabla} f(y_k, w_k) \Big| y_k \right] - \nabla f(y_k)^\top \nabla f(y_k) \right] \qquad (28)$$

$$\leq \mathbb{E}\left[ (A\tilde{\xi}_k + B\nabla f(y_k))^\top P(A\tilde{\xi}_k + B\nabla f(y_k)) \right] + \sigma^2 \alpha^2 \tilde{P}_{11} \qquad (29)$$

$$= \mathbb{E}\left[ \begin{bmatrix} \tilde{\xi}_k \\ \nabla f(y_k) \end{bmatrix}^\top \begin{bmatrix} A^\top PA & A^\top PB \\ B^\top PA & B^\top PB \end{bmatrix} \begin{bmatrix} \tilde{\xi}_k \\ \nabla f(y_k) \end{bmatrix} \right] + \sigma^2 \alpha^2 \tilde{P}_{11}. \qquad (30)$$

where in (27) we used the equality in (3) and the facts that we mentioned above. Also, (28) comes from the fact that $B^\top PB = \alpha^2 \tilde{P}_{11} I_d$ which can be shown by substituting $B$ from (9) and using the assumption $P = \tilde{P} \otimes I_d$. Finally (29) follows from the inequality in (3), and (30) is obtained by writing the first term of (29) in matrix format. □

Similarly, by extending Lemma 4.5 in [2] to the noise setting (3), for every $k \geq 0$ we obtain

$$\mathbb{E}\left[ \begin{bmatrix} \xi_k - \xi^* \\ \nabla f(y_k) \end{bmatrix}^\top \left( \rho^2 X_1 + (1 - \rho^2) X_2 \right) \begin{bmatrix} \xi_k - \xi^* \\ \nabla f(y_k) \end{bmatrix} \right] \leq \rho^2 \mathbb{E}[f(x_k) - f^*] - \mathbb{E}[f(x_{k+1}) - f^*]$$

$$+ \frac{L\alpha^2}{2} \sigma^2$$

where $X_1 = \tilde{X}_1 \otimes I_d$ and $X_2 = \tilde{X}_2 \otimes I_d$. The rest of the proof of Lemma 2.2 is very similar to the proof of Theorem 4.6 in [2], and we just need to use the fact that the Kronecker product of two positive semidefinite matrices is positive semidefinite [10].

## C   Proof of Theorem 2.3

Let

$$\Gamma \triangleq \rho^2 \tilde{X}_1 + (1 - \rho^2)\tilde{X}_2 - \begin{bmatrix} \tilde{A}^\top \tilde{P} \tilde{A} - \rho^2 \tilde{P} & \tilde{A}^\top \tilde{P} \tilde{B} \\ \tilde{B}^\top \tilde{P} \tilde{A} & \tilde{B}^\top \tilde{P} \tilde{B} \end{bmatrix}$$

with $\rho^2 = 1 - \sqrt{\alpha\mu}$ and $P = \tilde{P}_\alpha \otimes I_d$. According to Lemma 2.2, it suffices to show that $\Gamma \succeq 0_3$. Using the Symbolic toolbox in MATLAB, we see that $\Gamma$ has the following properties

  (i)  $\Gamma_{3,3} = \dfrac{\alpha(1 - L\alpha)}{2}$,

  (ii)  $\Gamma_{2,3} = \Gamma_{3,2} = 0$,

(iii) $\Gamma_{2,2} = \dfrac{\mu\sqrt{\mu\alpha}(1-\sqrt{\mu\alpha})^2}{2(1+\sqrt{\mu\alpha})^2} + \dfrac{\mu(1-\sqrt{\mu\alpha})^3}{2(1+\sqrt{\mu\alpha})^2} - \dfrac{(1-\sqrt{\mu\alpha})^2}{2\alpha(1+\sqrt{\mu\alpha})^2} + \dfrac{(1-\sqrt{\mu\alpha})^3}{2\alpha}$

$\qquad = \dfrac{(1-\sqrt{\mu\alpha})^2}{2\alpha(1+\sqrt{\mu\alpha})^2}\left(\alpha\Big(\mu\sqrt{\mu\alpha}+\mu(1-\sqrt{\mu\alpha})\Big)-1+(1-\sqrt{\mu\alpha})(1+\sqrt{\mu\alpha})^2\right)$

$\qquad = \dfrac{(1-\sqrt{\alpha\mu})^3\sqrt{\mu}}{2\sqrt{\alpha}(1+\sqrt{\alpha\mu})} \geq 0$

(iv) $\det(\Gamma) = 0$.

In fact, if $\alpha = \frac{1}{L}$, then

$$\Gamma = \frac{(1-\sqrt{\alpha\mu})^3\sqrt{\mu}}{2\sqrt{\alpha}(1+\sqrt{\alpha\mu})}\begin{bmatrix} 1 & -1 & 0 \\ -1 & 1 & 0 \\ 0 & 0 & 0 \end{bmatrix},$$

which is positive semidefinite. Now, consider the case that $\alpha < \frac{1}{L}$. For any $\epsilon > 0$, let

$$\Gamma^\epsilon \triangleq \Gamma + \epsilon \begin{bmatrix} 1 & 0 & 0 \\ 0 & 0 & 0 \\ 0 & 0 & 0 \end{bmatrix}.$$

Note that, for any $\epsilon > 0$, $(i)$ implies $\Gamma^\epsilon_{3,3} > 0$. This fact, along with (ii) and (iii), indicates that $\det(\Gamma^\epsilon_{[2:3],[2:3]}) > 0$. Hence,

$$\det(\Gamma^\epsilon) = \det(\Gamma) + \epsilon \det(\Gamma^\epsilon_{[2:3],[2:3]}) = \epsilon \det(\Gamma^\epsilon_{[2:3],[2:3]}) > 0$$

where the second equality comes from (iv). Therefore, the determinant of $\Gamma^\epsilon$, itself, and two submatrices $\Gamma^\epsilon_{3,3}$ and $\Gamma^\epsilon_{[2:3],[2:3]}$ are all positive. Thus, by Sylvester's criterion, $\Gamma^\epsilon$ is positive definite for any $\epsilon > 0$. As a consequence, since $\Gamma = \lim_{\epsilon \to 0} \Gamma^\epsilon$, $\Gamma$ is positive semidefinite.

## D  Proof of Theorem 3.1

Using Theorem 2.3, for every $k \geq 1$, we have

$$\mathbb{E}\left[V_{P_\alpha}(\xi_{k+1})\right] \leq (1-\frac{c}{\sqrt{\kappa}})\mathbb{E}\left[V_{P_\alpha}(\xi_k)\right] + \frac{\sigma^2}{2L}c^2(1+c^2)$$

$$\leq (1-\frac{c}{\sqrt{\kappa}})\mathbb{E}\left[V_{P_\alpha}(\xi_k)\right] + \frac{\sigma^2}{L}c^2$$

where in the last inequality we used the fact that $c^2 \leq 1$. Using this bound recursively for $n$ times, we obtain

$$\mathbb{E}\left[V_{P_\alpha}(\xi_{n+1})\right] \leq (1-\frac{c}{\sqrt{\kappa}})^n \mathbb{E}\left[V_{P_\alpha}(\xi_1)\right] + \frac{\sigma^2}{L}c^2 \sum_{i=1}^{n}(1-\frac{c}{\sqrt{\kappa}})^{n-i}$$

$$\leq \exp(-n\frac{c}{\sqrt{\kappa}})\mathbb{E}\left[V_{P_\alpha}(\xi_1)\right] + \frac{\sigma^2}{L}c^2 \frac{1-(1-\frac{c}{\sqrt{\kappa}})^n}{1-(1-\frac{c}{\sqrt{\kappa}})}$$

$$\leq \exp(-n\frac{c}{\sqrt{\kappa}})\mathbb{E}\left[V_{P_\alpha}(\xi_1)\right] + \frac{\sigma^2}{L}c\sqrt{\kappa}$$

where the second inequality follows from the inequality that $1-t \leq \exp(-t)$ for every $t \geq 0$, and the third inequality is obtained by replacing $1-(1-\frac{c}{\sqrt{\kappa}})^n$ by 1.

## E  Proof of Corollary 3.2

We first show $\alpha_1 \leq \frac{1}{L}$. Note that, by assumption, $n$ can be written as $p\sqrt{\kappa}n_0$ where $n_0 \geq 2\log(p\sqrt{\kappa})$ and $n_0 \geq e$. This assumption, along with the fact that $\frac{\log n}{n}$ is a decreasing function of $n$ as $n \geq e$,

implies

$$\frac{p\sqrt{\kappa}\log n}{n} = \frac{p\sqrt{\kappa}\log(p\sqrt{\kappa}n_0)}{p\sqrt{\kappa}n_0}$$

$$= \frac{\log(p\sqrt{\kappa}) + \log n_0}{n_0}$$

$$\leq \frac{1}{2} + \frac{\log n_0}{n_0} \tag{31}$$

$$\leq 1 \tag{32}$$

where in (31) we used the assumption $n_0 \geq 2\log(p\sqrt{\kappa})$, and (32) follows from the fact that $n_0 \geq e$, and therefore, $\frac{\log n_0}{n_0} \leq 1/e \leq 1/2$.

Next, using Theorem 3.1 with $c = \frac{p\sqrt{\kappa}\log n}{n}$ immediately gives the desired bound.

## F   Proof of Lemma 3.3

First, note that $\xi_1^{k+1} = \left[{x_{n_k+1}^k}^\top, {x_{n_k+1}^k}^\top\right]^\top$, and therefore,

$$(\xi_1^{k+1} - \xi^*)^\top P_{\alpha_{k+1}}(\xi_1^{k+1} - \xi^*)$$

$$= (\xi_1^{k+1} - \xi^*)^\top \begin{bmatrix} \sqrt{\frac{1}{2\alpha_{k+1}}} \\ \sqrt{\frac{\mu}{2}} - \sqrt{\frac{1}{2\alpha_{k+1}}} \end{bmatrix} \begin{bmatrix} \sqrt{\frac{1}{2\alpha_{k+1}}} & \sqrt{\frac{\mu}{2}} - \sqrt{\frac{1}{2\alpha_{k+1}}} \end{bmatrix} (\xi_1^{k+1} - \xi^*)$$

$$= \|x_{n_k+1}^k - x^*\|^2 \left(\frac{1}{2\alpha_{k+1}} + (\sqrt{\frac{\mu}{2}} - \sqrt{\frac{1}{2\alpha_{k+1}}})^2 + \frac{\sqrt{2}}{\sqrt{\alpha_{k+1}}}(\sqrt{\frac{\mu}{2}} - \sqrt{\frac{1}{2\alpha_{k+1}}})\right)$$

$$= \frac{\mu}{2}\|x_{n_k+1}^k - x^*\|^2. \tag{33}$$

Plugging (33) into (10) for $V_{P_{\alpha_{k+1}}}(\xi_1^{k+1})$ yields

$$V_{P_{\alpha_{k+1}}}(\xi_1^{k+1}) = (\xi_1^{k+1} - \xi^*)^\top P_{\alpha_{k+1}}(\xi_1^{k+1} - \xi^*) + f(x_{n_k+1}^k) - f^*$$

$$= \frac{\mu}{2}\|x_{n_k+1}^k - x^*\|^2 + f(x_{n_k+1}^k) - f^*$$

$$\leq 2(f(x_{n_k+1}^k) - f^*) \tag{34}$$

$$\leq 2V_{P_{\alpha_k}}(\xi_{n_k+1}^k) \tag{35}$$

where (34) follows from (2) with $x = x_{n_k+1}^k$ and $y = x^*$. Finally, taking expectation from (35) completes the proof.

## G   Proof of Theorem 3.4

We claim that for every $k \geq 1$

$$\mathbb{E}\left[V_{P_{\alpha_k}}(\xi_{n_k+1}^k)\right] \leq \frac{2}{2^{(p+1)(k-1)}}\left(\exp(-n_1/\sqrt{\kappa})(f(x_0^0) - f^*)\right) + \frac{\sigma^2\sqrt{\kappa}}{L2^{k-1}}. \tag{36}$$

which implies (17) as $V_{P_{\alpha_k}}(\xi_{n_k+1}^k) \geq f(x_{n_k+1}^k) - f^*$. We show (36) by induction on $k$. For $k = 1$, using Theorem 3.1, we obtain

$$\mathbb{E}\left[V_{P_{\alpha_1}}(\xi_{n_1+1}^1)\right] \leq \exp(-\frac{n_1}{\sqrt{\kappa}})\mathbb{E}\left[V_{P_{\alpha_1}}(\xi_1)\right] + \frac{\sigma^2\sqrt{\kappa}}{L}$$

$$\leq 2\exp(-\frac{n_1}{\sqrt{\kappa}})(f(x_0^0) - f^*) + \frac{\sigma^2\sqrt{\kappa}}{L} \tag{37}$$

where the second inequality comes from the inequality $V_{P_{\alpha_1}}(\xi_1) \leq 2(f(x_0^0) - f^*)$ which can be derived similar to (34).

Next, we assume (36) holds for $k$ and show it also holds for $k + 1$. Note that

$$\mathbb{E}\left[V_{P_{\alpha_{k+1}}}(\xi_{n_{k+1}+1}^{k+1})\right] \leq \exp\left(-n_{k+1}\sqrt{\frac{\alpha_{k+1}L}{\kappa}}\right)\mathbb{E}\left[V_{P_{\alpha_{k+1}}}(\xi_1^{k+1})\right] + \frac{\sigma^2\sqrt{\kappa}\sqrt{\alpha_{k+1}L}}{L} \quad (38)$$

$$= \exp\left(-\log(2^{p+2})\right)\mathbb{E}\left[V_{P_{\alpha_{k+1}}}(\xi_1^{k+1})\right] + \frac{\sigma^2\sqrt{\kappa}}{2^{k+1}L}$$

$$\leq \frac{2}{2^{p+2}}\mathbb{E}\left[V_{P_{\alpha_k}}(\xi_{n_k+1}^k)\right] + \frac{\sigma^2\sqrt{\kappa}}{2^{k+1}L} \quad (39)$$

$$\leq \frac{1}{2^{p+1}} \cdot \frac{2}{2^{(p+1)(k-1)}}\left(\exp(-n_1/\sqrt{\kappa})(f(x_0^0) - f^*)\right) + \frac{\sigma^2\sqrt{\kappa}}{L}\left(\frac{1}{2^{k+1}} + \frac{1}{2^{p+k}}\right) \quad (40)$$

$$\leq \frac{2}{2^{(p+1)k}}\left(\exp(-n_1/\sqrt{\kappa})(f(x_0^0) - f^*)\right) + \frac{\sigma^2\sqrt{\kappa}}{L2^k} \quad (41)$$

where, (38) and (39) are obtained by using Theorem 3.1 and Lemma 3.3, respectively, and in (40) we used the assumption that (36) holds for $k$.

## H  Proof of Theorem 3.6

First, we will show that for every $k \geq 1$ and $0 \leq m \leq n_k + 1$, we have

$$\mathbb{E}\left[f(x_m^{k+1})\right] - f^* \leq \frac{4}{2^{(p+1)(k-1)}}\left(\exp(-n_1/\sqrt{\kappa})(f(x_0^0) - f^*)\right) + \frac{9\sigma^2\sqrt{\kappa}}{4L2^{k-1}}. \quad (42)$$

Indeed,

$$\mathbb{E}\left[f(x_m^{k+1})\right] - f^* \leq \mathbb{E}\left[V_{P_{\alpha_{k+1}}}(\xi_m^{k+1})\right]$$

$$\leq \exp\left(-m\sqrt{\frac{\alpha_{k+1}L}{\kappa}}\right)\mathbb{E}\left[V_{P_{\alpha_{k+1}}}(\xi_1^{k+1})\right] + \frac{\sigma^2\sqrt{\kappa}\sqrt{\alpha_{k+1}L}}{L} \quad (43)$$

$$\leq 2\mathbb{E}\left[V_{P_{\alpha_k}}(\xi_{n_k+1}^k)\right] + \frac{\sigma^2\sqrt{\kappa}}{2^{k+1}L} \quad (44)$$

$$\leq \frac{4}{2^{(p+1)(k-1)}}\left(\exp(-n_1/\sqrt{\kappa})(f(x_0^0) - f^*)\right) + \frac{9\sigma^2\sqrt{\kappa}}{4L2^{k-1}} \quad (45)$$

where, (43) and (44) follows again from Theorem 3.1 and Lemma 3.3, and we obtain (45) using (36).

Recall the definition $N_K(p, n_1) \triangleq \sum_{k=1}^{K} n_k$ which denotes the total number of stochastic gradient iterations required to complete $K$ stages of M-ASG for parameter $p$ and first-stage iteration number $n_1$ fixed. Given the computational budget of $n$ iterations such that $n \geq n_1$, let $K$ be the largest number such that $n \geq N_K \triangleq N_K(p, n_1)$. As a result, at iteration $n$ we are in stage $K + 1$. Note that (18) implies $2^{K-1} \geq \frac{N_K - n_1 + \Psi}{\Psi}$ with $\Psi = 4\lceil(p + 2)\sqrt{\kappa}\log(2)\rceil$; therefore

$$\frac{1}{2^{K-1}} \leq \frac{\Psi}{N_K - n_1 + \Psi}. \quad (46)$$

Thus, we get the following upper bound on the suboptimality:

$$\mathbb{E}\left[f(x_n)\right] - f^* = \mathbb{E}\left[f(x_{n-N_K}^{K+1})\right] - f^*$$

$$\leq \frac{4}{2^{(p+1)(K-1)}}\left(\exp(-n_1/\sqrt{\kappa})(f(x_0^0) - f^*)\right) + \frac{9\sigma^2\sqrt{\kappa}}{4L2^{K-1}}, \quad (47)$$

and by substituting (46) in (47) we obtain the bound

$$\mathbb{E}\left[f(x_n)\right] - f^*$$

$$\leq \mathcal{O}(1)\left(\frac{(4(p+1)\sqrt{\kappa}\log(2))^{p+1}}{(N_K - n_1 + \Psi)^{p+1}}\left(\exp(-n_1/\sqrt{\kappa})(f(x_0^0) - f^*)\right) + \frac{(p+1)\sigma^2}{(N_K - n_1 + \Psi)\mu}\right). \quad (48)$$

Next, by (18), $N_{K+1} - n_1 \leq 2(N_K - n_1 + \Psi)$, and thus, $n - n_1 \leq 2(N_K - n_1 + \Psi)$. Replacing $\frac{1}{N_K - n_1 + \Psi}$ by $\frac{2}{n - n_1}$ in (48) completes the proof of (19).

# I Proof of Corollary 3.7

Note that by setting $n_1 = \lceil (p+1)\sqrt{\kappa}\log(12(p+1)\kappa)\rceil$, we have $\exp(-n_1/\sqrt{\kappa}) \leq \frac{1}{(16\log(2)(p+1)\sqrt{\kappa})^{p+1}}$; hence, plugging $n_1 = \lceil (p+1)\sqrt{\kappa}\log(12(p+1)\kappa)\rceil$ in (19) implies the following bound with an $\mathcal{O}(1)$ constant that does not depend on $\mu$, $L$ and $x_0^0$:

$$\mathcal{O}(1)\left(\frac{2^{-(p+1)}}{(n-n_1)^{p+1}}(f(x_0) - f^*) + \frac{(p+1)\sigma^2}{(n-n_1)\mu}\right). \tag{49}$$

Finally, note that $n \geq 2n_1$; therefore, $n - n_1 \geq n/2$ and using it in (49) completes the proof.

# J Proof of Corrolary 3.9

By plugging $p = 1$ and $n_1 = \lceil \sqrt{\kappa}\log\left(\frac{4\Delta}{\epsilon}\right)\rceil$ in (17), it is straightforward to check the bias term is bounded by $\frac{\epsilon}{2}$. Next, consider running M-ASG with given parameters, possibly without knowing and/or specifying the exact number of stages. Consider the end of the $K$-th stage, where $K \triangleq \lceil \log_2(\frac{\sigma^2\sqrt{\kappa}}{L\epsilon})\rceil + 2$. Since $\frac{1}{2^{K-1}} \leq \frac{L\epsilon}{2\sigma^2\sqrt{\kappa}}$, the variance term in (17) is also bounded by $\frac{\epsilon}{2}$, and as a result $x_{n_K+1}^K$ is an $\epsilon$−solution.

Now, by using (18), we can bound the number of iterations for completing $K$ stages:

$$N_K = n_1 + (2^{K+1} - 4)\lceil\sqrt{\kappa}\log(8)\rceil \leq n_1 + 2(1 + \log(8))\left(2^K\sqrt{\kappa}\right) \tag{50}$$

$$\leq \lceil\sqrt{\kappa}\log\left(\frac{4\Delta}{\epsilon}\right)\rceil + 16(1 + \log(8))\frac{\sigma^2}{\mu\epsilon} \tag{51}$$

where in (50) we used the fact that $\lceil\sqrt{\kappa}\log(8)\rceil \leq (1 + \frac{1}{\log(8)})\sqrt{\kappa}\log(8)$ since $\kappa \geq 1$, and (51) follows from the bound $K \leq 3 + \log_2(\frac{\sigma^2\sqrt{\kappa}}{L\epsilon})$.

# K Results for More General Noise Setting

In this section, we show how our analysis can be extended to a more general noise setting where the bound on the variance can depend on the distance to the optimal solution. More formally, we assume that at $x \in \mathbb{R}^d$, we have access to the noisy gradient $\tilde{\nabla}f(x,w)$ such that for some $\sigma, \eta \geq 0$,

$$\begin{aligned} \mathbb{E}[\tilde{\nabla}f(x,w)|x] &= \nabla f(x) \\ \mathbb{E}\left[\|\tilde{\nabla}f(x,w) - \nabla f(x)\|^2 \Big| x\right] &\leq \sigma^2 + \eta^2\|x - x^*\|^2, \end{aligned} \tag{52}$$

where $w$ is a random variable independent of previous iterates.

In what follows, we first show how the results of Theorem 2.3 and Lemma 3.3 extends to this setting, and then briefly discuss the results of our multistage scheme for this noise setting.

**Theorem K.1.** *Let $f \in S_{\mu,L}(\mathbb{R}^d)$ with $\kappa \geq 4$. Consider the ASG iterations given in (7) under noise model in (52). For $\alpha \in (0, \bar{\alpha}]$ and $\beta = \frac{1 - \sqrt{\alpha\mu}}{1 + \sqrt{\alpha\mu}}$ with*

$$\bar{\alpha} := \begin{cases} \min\{\frac{1}{L}, \frac{\mu^3}{(60\eta^2)^2}\} & \text{if } \eta > 0, \\ \frac{1}{L} & \text{if } \eta = 0, \end{cases} \tag{53}$$

*it follows that*

$$\mathbb{E}\left[V_{Q_\alpha}(\xi_{k+1})\right] \leq (1 - \sqrt{\alpha\mu}/3)\mathbb{E}\left[V_{Q_\alpha}(\xi_k)\right] + 2\sigma^2\alpha, \tag{54}$$

*for every $k \geq 0$, where $Q_\alpha = \tilde{Q}_\alpha \otimes I_d$ with*

$$\tilde{Q}_\alpha = \tilde{P}_\alpha + 2\alpha\eta^2\tilde{C}^\top\tilde{C} = \begin{bmatrix} \sqrt{\frac{1}{2\alpha}} \\ \sqrt{\frac{\mu}{2}} - \sqrt{\frac{1}{2\alpha}} \end{bmatrix}\begin{bmatrix} \sqrt{\frac{1}{2\alpha}} & \sqrt{\frac{\mu}{2}} - \sqrt{\frac{1}{2\alpha}} \end{bmatrix} + 2\alpha\eta^2\begin{bmatrix} 1 + \beta \\ -\beta \end{bmatrix}[1 + \beta \quad -\beta].$$

*Proof.* First, note that similar to the proof of Lemma 2.2 and by using $\alpha L \leq 1$, we can show

$$\mathbb{E}\left[V_{P_\alpha}(\xi_{k+1})\right] \leq (1 - \sqrt{\alpha\mu})\mathbb{E}\left[V_{P_\alpha}(\xi_k)\right] + \alpha\sigma^2 + \alpha\eta^2\mathbb{E}\left[\|y_k - x^*\|^2\right]. \tag{55}$$

Using $y_k = C\xi_k$, we can substitute $\|y_k - x^*\|^2$ by $(\xi_k - \xi^*)^\top C^\top C(\xi_k - \xi^*)$ in (55); hence,

$$\mathbb{E}\left[V_{P_\alpha}(\xi_{k+1})\right] \leq (1 - \sqrt{\alpha\mu})\mathbb{E}\left[(\xi_k - \xi^*)\right] + \frac{1}{2}\mathbb{E}\left[(\xi_k - \xi^*)^\top 2\alpha\eta^2 C^\top C(\xi_k - \xi^*)\right] + \alpha\sigma^2$$

$$\leq (1 - \sqrt{\alpha\mu})\mathbb{E}\left[V_{Q_\alpha}(\xi_k)\right] + \alpha\sigma^2, \tag{56}$$

where the last inequality follows from $1 - \sqrt{\alpha\mu} \geq 1/2$ which is true since $\alpha \leq 1/L$ and $\kappa \geq 4$. Also note that

$$(\xi_k - \xi^*)^\top C^\top C(\xi_k - \xi^*) = \|(1 + \beta)(x_{k+1} - x^*) - \beta(x_k - x^*)\|^2$$

$$\leq 2(1 + \beta)^2\|x_{k+1} - x^*\|^2 + 2\beta^2\|x_k - x^*\|^2 \tag{57}$$

$$\leq \frac{16}{\mu}\left(f(x_{k+1}) - f^*\right) + \frac{4}{\mu}\left(f(x_k) - f^*\right) \tag{58}$$

$$\leq \frac{16}{\mu}V_{P_\alpha}(\xi_{k+1}) + \frac{4}{\mu}V_{Q_\alpha}(\xi_k). \tag{59}$$

where (57) follows from $(a + b)^2 \leq 2a^2 + 2b^2$ and (58) follows from $\beta \leq 1$ and the strong convexity assumption, i.e., $f(x) - f^* \geq \frac{\mu}{2}\|x - x^*\|^2$. Finally, (59) is obtained using $\min\{V_{P_\alpha}(\xi_k), V_{Q_\alpha}(\xi_k)\} \geq f(x_k) - f^*$. Plugging (59) into the definition of $V_{Q_\alpha}(\xi_{k+1})$ implies

$$\mathbb{E}\left[V_{Q_\alpha}(\xi_{k+1})\right] = \mathbb{E}\left[V_{P_\alpha}(\xi_{k+1})\right] + 2\alpha\eta^2\mathbb{E}\left[(\xi_k - \xi^*)^\top C^\top C(\xi_k - \xi^*)\right]$$

$$\leq (1 + \frac{32\alpha\eta^2}{\mu})\mathbb{E}\left[V_{P_\alpha}(\xi_{k+1})\right] + \frac{8\alpha\eta^2}{\mu}V_{Q_\alpha}(\xi_k). \tag{60}$$

Using this result along with (56) yields

$$\mathbb{E}\left[V_{Q_\alpha}(\xi_{k+1})\right] \leq \left((1 - \sqrt{\alpha\mu})(1 + \frac{32\alpha\eta^2}{\mu}) + \frac{8\alpha\eta^2}{\mu}\right)\mathbb{E}\left[V_{Q_\alpha}(\xi_k)\right] + \alpha(1 + \frac{32\alpha\eta^2}{\mu})\sigma^2$$

$$\leq \left(1 - \sqrt{\alpha\mu} + \frac{40\alpha\eta^2}{\mu}\right)\mathbb{E}\left[V_{Q_\alpha}(\xi_k)\right] + \alpha(1 + \frac{32\alpha\eta^2}{\mu})\sigma^2$$

$$\leq (1 - \sqrt{\alpha\mu}/3)\mathbb{E}\left[V_{Q_\alpha}(\xi_k)\right] + (1 + \frac{32\alpha\eta^2}{\mu})\alpha\sigma^2 \tag{61}$$

where the last inequality follows from the assumption $\alpha \leq \frac{\mu^3}{(60\eta^2)^2}$ which implies

$$\frac{2}{3}\sqrt{\alpha\mu} \geq \frac{40\alpha\eta^2}{\mu}. \tag{62}$$

Finally, note that, (62) along with $\alpha \leq 1/L$ also implies $1 \geq 60\alpha\eta^2/\mu$; thus, we can bound $1 + 32\alpha\eta^2/\mu$ in (61) by 2 which gives us the desired result. $\qquad\square$

Next, note that we can also extend Lemma 3.3 to the new Lyapunov function $\mathbb{E}\left[V_{Q_\alpha}(\xi_k)\right]$ as well:

**Lemma K.2.** *Let* $f \in S_{\mu,L}(\mathbb{R}^d)$. *Consider M-ASG, i.e., Algorithm 1, assuming the noise model in* (52), *with* $\alpha \in (0, \bar{\alpha}]$, *where* $\bar{\alpha}$ *is defined by* (53). *Then, for every* $1 \leq k \leq K - 1$,

$$\mathbb{E}\left[V_{Q_{\alpha_{k+1}}}(\xi_1^{k+1})\right] \leq 3\mathbb{E}\left[V_{Q_{\alpha_k}}(\xi_{n_k+1}^k)\right]. \tag{63}$$

*Proof.* The proof is very similar to the arguments in Appendix F. In particular,

$$V_{Q_{\alpha_{k+1}}}(\xi_1^{k+1}) = (\xi_1^{k+1} - \xi^*)^\top Q_{\alpha_{k+1}}(\xi_1^{k+1} - \xi^*) + f(x_{n_k+1}^k) - f^*$$

$$= \frac{\mu}{2}\|x_{n_k+1}^k - x^*\|^2 + 2\alpha\eta^2(\xi_1^{k+1} - \xi^*)^\top C^\top C(\xi_1^{k+1} - \xi^*) + f(x_{n_k+1}^k) - f^*$$

$$= (\frac{\mu}{2} + 2\alpha\eta^2)\|x_{n_k+1}^k - x^*\|^2 + f(x_{n_k+1}^k) - f^*$$

$$\leq (2 + \frac{4\alpha\eta^2}{\mu})(f(x_{n_k+1}^k) - f^*) \tag{64}$$

$$\leq 3V_{Q_{\alpha_k}}(\xi_{n_k+1}^k) \tag{65}$$

where (64) follows from (2) with $x = x_{n_k+1}^k$ and $y = x^*$. Finally, (65) follows from $1/60 \geq \alpha \eta^2/\mu$, which holds due to (62) and $\alpha \leq \frac{1}{L}$, along with $V_{Q_{\alpha_k}}(\xi_{n_k+1}^k) \geq f(x_{n_k+1}^k) - f^*$. Taking expectations of both sides of (65) completes the proof. $\qquad \square$

Using the results in Theorem K.1 and Lemma K.2, we can analyze M-ASG for this more general noise setting in (52) as well and extend our complexity result in Corollary 3.8 as follows:

$$\mathbb{E}\left[f(x_n)\right] - f^* \leq \mathcal{O}(1)\left(\exp\left(-n/\left(\Theta(1)(\sqrt{\kappa} + \eta^2/\mu^2)\right)\right)(f(x_0^0) - f^*) + \frac{\sigma^2}{n\mu}\right)$$

for $n$ sufficiently large and known in advance. It is worth noting that we can also derive similar results to Theorems 3.4 and 3.6 when $n$ is not known. We skip the details as all the arguments follow very similar to our analysis in Section 3.

## L  M-ASG for Convex Objective Functions

For merely convex objective functions, as discussed in [22], the suboptimality $\mathbb{E}\left[f(x_n)\right] - f^*$ admits the lower bound given below:

$$\Theta(1)\left(\frac{L}{n^2}\|x_0 - x^*\|_2^2 + \frac{\sigma^2}{\sqrt{n}}\right). \tag{66}$$

The author of [22] obtains this lower bound for the case of compact domain with the additional knowledge of noise parameter $\sigma$. For unconstrained optimization, and without using the information on the noise parameter, $\sigma^2$, it is shown in [8] that one can achieve the rate $\mathcal{O}(\frac{1}{\sqrt{n}})$ in both bias and variance terms (see last part of Corollary 3.9 and also Corollary 4.1 in [8]). As we state below, a direct application of our current results recovers a similar result up to a log factor.

**Theorem L.1.** *Let $f$ be a merely convex function, i.e., $f \in S_{0,L}(\mathbb{R}^d)$ with $\mu = 0$, and let $n \geq 2$ be the given iteration budget. Define $f_\lambda(x) \triangleq f(x) + \frac{\lambda}{2}\|x - x_0\|^2$ with $\lambda \triangleq L/(\sqrt{n} - 1)$. Consider running ASG, given in (7), with stepsize $\alpha = \frac{(\log n)^2}{n^{3/2}L}$ for solving $\min_x f_\lambda(x)$. Then,*

$$\mathbb{E}\left[f(x_{n+1})\right] - f^* \leq \frac{2}{n}(f(x_0) - f^*) + \frac{L}{\sqrt{n}}\|x_0 - x^*\|^2 + \frac{\sigma^2 \log n}{\sqrt{n}L}. \tag{67}$$

*Proof.* Define $f_\lambda^* \triangleq \min_x f_\lambda(x)$. Note $f_\lambda \in S_{\lambda, L+\lambda}(\mathbb{R}^d)$; thus, using Theorem 3.1 with $c = \log n/n^{3/4}$ and $\kappa = (L + \lambda)/\lambda = \sqrt{n}$ implies

$$
\begin{aligned}
\mathbb{E}\left[f_\lambda(x_{n+1}^1)\right] - f_\lambda^* &\leq \mathbb{E}\left[V_{P_\alpha}(\xi_{n+1})\right] \\
&\leq \exp(-n\frac{c}{\sqrt{\kappa}})\mathbb{E}\left[V_{P_\alpha}(\xi_1)\right] + \frac{\sigma^2\sqrt{\kappa}c}{L + \lambda} \\
&\leq \frac{1}{n}\mathbb{E}\left[V_{P_\alpha}(\xi_1)\right] + \frac{\sigma^2 \log n}{\sqrt{n}L}. \tag{68}
\end{aligned}
$$

Now, using the fact that $x_0 = x_{-1}$, and similar to the proof of Lemma 3.3, we can show

$$\mathbb{E}[V_{P_\alpha}(\xi_1)] \leq 2(f_\lambda(x_0) - f_\lambda^*) = 2(f(x_0) - f_\lambda^*).$$

Therefore, plugging this into (68), we obtain

$$\mathbb{E}\left[f_\lambda(x_{n+1})\right] - f_\lambda^* \leq \frac{2}{n}(f(x_0) - f_\lambda^*) + \frac{\sigma^2 \log n}{\sqrt{n}L}, \tag{69}$$

which is equivalent to

$$\mathbb{E}\left[f_\lambda(x_{n+1})\right] - (1 - \frac{2}{n})f_\lambda^* \leq \frac{2}{n}f(x_0) + \frac{\sigma^2 \log n}{\sqrt{n}L}. \tag{70}$$

This result along with $f(x_{n+1}) \leq f_\lambda(x_{n+1})$ implies

$$\mathbb{E}[f(x_{n+1})] - (1 - \frac{2}{n})f_\lambda^* \leq \frac{2}{n}f(x_0) + \frac{\sigma^2 \log n}{\sqrt{n}L}. \tag{71}$$

Finally, using the bound

$$f_\lambda^* \leq f_\lambda(x^*) = f^* + \frac{\lambda}{2}\|x_0 - x^*\|^2$$

completes the proof. □

# M   AC-SA from the perspective of Nesterov's Accelerated Method

Recall that AC-SA [16] with initial point $x_0$ and sequence of stepsize parameters $\{\eta_t\}_{t\geq 1}$ and $\{\gamma_t\}_{t\geq 1}$ has the following update rule:

(i) Set $x_0^{ag} = x_0$ and $t = 1$;

(ii) Set $x_t^{md} = \frac{(1-\eta_t)(\mu+\gamma_t)x_{t-1}^{ag}+\eta_t[(1-\eta_t)\mu+\gamma_t]x_{t-1}}{\gamma_t+(1-\eta_t^2)\mu}$;

(iii) Set $x_t = \frac{\eta_t\mu x_t^{md}+[(1-\eta_t)\mu+\gamma_t]x_{t-1}-\eta_t G_t}{\mu+\gamma_t}$ where $G_t = \tilde{\nabla} f(x_t^{md}, w_t)$;

(iv) Set $x_t^{ag} = \eta_t x_t + (1 - \eta_t)x_{t-1}^{ag}$;

(v) Set $t \leftarrow t + 1$ and go to step (ii).

We claim that this algorithm can be cast as an ASG method in (7) with a specific varying stepsize rule. In fact, we show it can be represented as

$$x_t^{md} = (1 + \tilde{\beta}_t)x_{t-1}^{ag} - \tilde{\beta}_t x_{t-2}^{ag} \tag{72a}$$

$$x_t^{ag} = x_t^{md} - \tilde{\alpha}_t \tilde{\nabla} f(x_t^{md}, w_t). \tag{72b}$$

with

$$\tilde{\alpha}_t = \frac{\eta_t^2}{\mu + \gamma_t}, \quad \tilde{\beta}_t = \frac{\eta_t(1 - \eta_{t-1})[(1 - \eta_t)\mu + \gamma_t]}{\eta_{t-1}[\gamma_t + (1 - \eta_t^2)\mu]}.$$

To show this, first, multiplying both sides of ((ii)) by $\frac{\gamma_t+(1-\eta_t^2)\mu}{\mu+\gamma_t}$ implies

$$\frac{\gamma_t + (1 - \eta_t^2)\mu}{\mu + \gamma_t}x_t^{md} = (1 - \eta_t)x_{t-1}^{ag} + \frac{\eta_t[(1 - \eta_t)\mu + \gamma_t]}{\mu + \gamma_t}x_{t-1}, \tag{73}$$

and by substituting $(1 - \eta_t)x_{t-1}^{ag}$ by $x_t^{ag} - \eta_t x_t$ from ((iv)) we obtain

$$x_t^{ag} = \eta_t\left(x_t - \frac{(1 - \eta_t)\mu + \gamma_t}{\mu + \gamma_t}x_{t-1}\right) + \frac{\gamma_t + (1 - \eta_t^2)\mu}{\mu + \gamma_t}x_t^{md}. \tag{74}$$

Note that, by ((iii)), we have

$$x_t - \frac{(1 - \eta_t)\mu + \gamma_t}{\mu + \gamma_t}x_{t-1} = \frac{\eta_t\mu}{\mu + \gamma_t}x_t^{md} - \frac{\eta_t}{\mu + \gamma_t}G_t, \tag{75}$$

and therefore, (74) and (75) yield

$$\begin{aligned}
x_t^{ag} &= \eta_t\left(\frac{\eta_t\mu}{\mu + \gamma_t}x_t^{md} - \frac{\eta_t}{\mu + \gamma_t}G_t\right) + \frac{\gamma_t + (1 - \eta_t^2)\mu}{\mu + \gamma_t}x_t^{md} \\
&= \left(\frac{\eta_t^2\mu}{\mu + \gamma_t} + \frac{\gamma_t + (1 - \eta_t^2)\mu}{\mu + \gamma_t}\right)x_t^{md} - \frac{\eta_t^2}{\mu + \gamma_t}G_t \\
&= x_t^{md} - \frac{\eta_t^2}{\mu + \gamma_t}G_t
\end{aligned}$$

which implies (72b).

fig1: $\sigma_n^2 = 10^{-2}$     fig2: $\sigma_n^2 = 10^{-4}$     fig3: $\sigma_n^2 = 10^{-6}$

Figure 4: Comparison of GD, AGD, $\mu$AGD+, and MASG for logistic regression with $n = 1000$ iterations with different level of noise.

fig1: $\sigma_n^2 = 10^{-2}$     fig2: $\sigma_n^2 = 10^{-4}$     fig3: $\sigma_n^2 = 10^{-6}$

Figure 5: Comparison of GD, AGD, $\mu$AGD+, and MASG for logistic regression with $n = 10000$ iterations with different level of noise.

To show (72a), first note that by ((iv)) for $t - 1$, we obtain $x_{t-1} = \frac{1}{\eta_{t-1}}(x_{t-1}^{ag} - (1 - \eta_{t-1})x_{t-2}^{ag})$. Plugging in this in ((ii)), leads to

$$
\begin{aligned}
x_t^{md} &= \frac{(1 - \eta_t)(\mu + \gamma_t)}{\gamma_t + (1 - \eta_t^2)\mu} x_{t-1}^{ag} + \frac{\eta_t[(1 - \eta_t)\mu + \gamma_t]}{\eta_{t-1}[\gamma_t + (1 - \eta_t^2)\mu]} \left( x_{t-1}^{ag} - (1 - \eta_{t-1})x_{t-2}^{ag} \right) \\
&= \left( \frac{(1 - \eta_t)(\mu + \gamma_t)}{\gamma_t + (1 - \eta_t^2)\mu} + \frac{\eta_t[(1 - \eta_t)\mu + \gamma_t]}{\eta_{t-1}[\gamma_t + (1 - \eta_t^2)\mu]} \right) x_{t-1}^{ag} - \frac{\eta_t(1 - \eta_{t-1})[(1 - \eta_t)\mu + \gamma_t]}{\eta_{t-1}[\gamma_t + (1 - \eta_t^2)\mu]} x_{t-2}^{ag} \\
&= \frac{\eta_{t-1}(1 - \eta_t)(\mu + \gamma_t) + \eta_t[(1 - \eta_t)\mu + \gamma_t]}{\eta_{t-1}[\gamma_t + (1 - \eta_t^2)\mu]} x_{t-1}^{ag} - \tilde{\beta}_t x_{t-2}^{ag} \\
&= \frac{\eta_{t-1}(1 - \eta_t)(\mu + \gamma_t) + \eta_t\eta_{t-1}[(1 - \eta_t)\mu + \gamma_t]}{\eta_{t-1}[\gamma_t + (1 - \eta_t^2)\mu]} x_{t-1}^{ag} + \tilde{\beta}_t(x_{t-1}^{ag} - x_{t-2}^{ag}) \\
&= x_{t-1}^{ag} + \tilde{\beta}_t(x_{t-1}^{ag} - x_{t-2}^{ag})
\end{aligned}
$$

which is (72a) and the proof is complete.

As a consequence, Multistage AC-SA is a variant of M-ASG Algorithm that has a different length $n_k$ for each stage $k \geq 1$ and employs a specific varying stepsize rule together with a different selection for the momentum parameter at each stage.

## N    Additional Numerical Experiments

In this section, we study another classification problem using logistic regression, but with a synthesized data. In particular, we generate a random matrix $M \in \mathbb{R}^{2000 \times 100}$ and a random vector $w \in \mathbb{R}^{100}$ and compute $y = \text{sign}(Mw)$ which is the vector that contains the sign of the inner product with the rows of $M$ and the vector $w$. Our goal is to recover $w$ by optimizing a regularized logistic

objective when the gradient of the loss function is corrupted with additive Gaussian noise. We compare M-ASG and M-ASG$^*$ with Standard GD, Standard AG, $\mu$AGD+ [8], and Multistage AC-SA [17]. We note that the condition number of the problem $\kappa \sim 1000$ for this problem. Figures 4– 5 illustrate the behavior of the algorithms for $n = 1000$ and $n = 10000$ iterations for the noise level $\sigma_n^2 \in \{10^{-6}, 10^{-4}, 10^{-2}\}$ as before. It can be seen that both M-ASG and M-ASG$^*$ usually start faster, and do not perform worse than other algorithms in different scenarios; moreover, they outperform other algorithms when the iteration budget is limited or the noise level is small. Furthermore, note that in the setting where the noise is large, M-ASG$^*$ behaves better than M-ASG, as it terminates the first stage earlier, which is helpful as the noise is large; hence, the variance becomes term dominant in the first stage just after a few iterations.