[Reviews · NeurIPS 2019]

Reviewer 1



**********After author response*************** I appreciated that the authors took the time to try to relax their noise assumption. The idea seems promising but I cannot change my score without seeing a real and complete proof. Yet the fact remains that, even if I will not fight for this paper, I think that it deserves a submission. *************************************************** Originality: the aim of the paper is very general but very important: how to accelerate SGD ? The algorithm is based on the accelerated gradient descent from Nesterov together with a multistage approach that does not seem new neither. Yet they reach optimal rates for the algorithm which is the principal aim of the paper. Quality and clarity: The aim of the paper is clear and as robustness of the acceleration of gradient descent with respect to noise is important, the result stated is quite strong. Moreover, even if the topic is quite technical and several attempts in the past have been made, the topic is well reviewed and documented. However, the paper lacks from some clarity, especially in Sections 2 and 3, where this is rather a succession of lemmas and theorems than a clear discussion about the difference with the previous works or the method employed. Significance: as I said in the previous Section (contribution), the result was already proven in more specific settings, and the significance of this paper may be more methodological than theoretical or experimental.

Reviewer 2



Originality: This paper provides a clear and deep analysis of a multi-stage accelerated SGD algorithm. The results show that the expected function value gap is bounded by an exponential decay term plus a sublinear decay term related to noise. They recover the deterministic case in the single stage and zero noise special case, while reaching the lower bound O(\sigma^2/n) in the noise term. The paper contains sufficient novel results and is competitive comparing with related work. In particular, the main results reveal how to choose the right time to switch from constant stepsize to decaying stepsize, a crucial choice for the overall performance of stochastic algorithms. Quality/Clarity: The paper is written with care. The proofs are correct, clear and concise. The results are well presented and highlighted. A detailed comparison is made with related papers. For example, the authors showed that AC-SA can be seen as M-SAG with specific varying stepsize rule. The technique of exponential decaying stepsize allows to avoid the requirement on the knowledge of \sigma. Significance: The paper brings new algorithm and new results in accelerated stochastic gradient methods. The results match the lower bound without requiring the knowledge of noise level and provide insight into parameter choice of the algorithm. The new algorithm also perform asymptotically the best in the reported experiments. Comments: 1, Should the function value gap in 132 be changed to the Lyapunov function? Also, where is Corollary 3.2 needed in the whole paper? 2, Typo in Equation (32). I read the response.

Reviewer 3



Compared with existing work [8], the contribution of this work is incremental. The key idea of proposed M-ASG is to use a exponentially decreased step size and increased number of iterations for each stage. In fact, this idea is similar to the results presented in [8]. More importantly, the proposed results only work for strongly convex objective. As discussed in [8], compared with generally convex objective, the accelerated method is much less sensitive to noisy gradient estimate if the objective is strongly convex. Therefore, more challenging and important problem is how to design an accelerated algorithm so that it is robust to noisy gradient and able to achieve the optimal rate. Compared with u-AGD [8], as showed in Figure 2 and 3, he proposed algorithm does not show clear difference and improvement in terms of empirical performance. After rebuttal: the authors' feedback partially address my questions and I have changed my score.

[Author Response · NeurIPS 2019]

We thank the reviewers for their careful consideration and their feedback, our replies are provided below. We hope the
reviewers will consider improving the scores based on our responses and the extensions we plan to include in the paper.
**Reviewer #1**: **Contributions of our work**: Our paper contributes to the understanding of first order methods and leads
to novel accelerated algorithms. Our algorithms (M-ASG and M-ASG*) not only lead to optimal iteration complexity but
also perform well in practice as illustrated by our experiments. Therefore, we believe our paper contributes to both theory
and practice of accelerated SGD methods. On the technical side, we first obtain a tight characterization of the trade-off
between bias and variance terms for a one-stage algorithm with constant stepsize. Building on this result and choosing the
stage length and stepsize carefully at each stage, we can achieve optimal iteration complexity through a simple multistage
algorithm without knowing the noise characteristics as opposed to previous approaches in the literature. **Clarity of**
**Sections 2 & 3**: We will move parts of the technical results to the appendix and add more high-level discussions about
our results for a smoother reading, thanks for the suggestion. **Relaxing our noise assumption**: Assumption H2 of Bach
& Moulines states that each unbiased estimate of gradient is Lipschitz. As a result, Assumptions H2 and H4 together
implies that there exist constants $\sigma_1, \sigma_2 > 0$ such that $\mathbb{E}[\|\tilde{\nabla} f(x_n, w_n) - \nabla f(x_n)\|^2 \mid x_n] \leq \sigma_1^2 + \sigma_2^2 \|x_n - x^*\|^2$. Our
analysis also extends to this noise model and we thank the reviewer for suggesting this. We will add a detailed section
in the appendix to elaborate on this. Here, due to the space limit, we explain the idea briefly: Note that Lemma 2.2
holds for this noise model as well if $\sigma^2$ is replaced by $\sigma_1^2 + \sigma_2^2 \mathbb{E}[\|y_k - x^*\|^2]$ because of the conditional expectation
technique that we use in the proof. Plugging $y_k = C\xi_k$, the result of Theorem 2.3 for $\alpha \leq 1/L$ will be replaced by
$\mathbb{E}\left[V_{P_\alpha}(\xi_{k+1})\right] \leq (1 - \sqrt{\alpha\mu})\mathbb{E}\left[V_{P_\alpha}(\xi_k)\right] + 2\sigma_1^2\alpha + \sigma_2^2\mathbb{E}[(\xi_k - \xi^*)^\top (C^\top C)(\xi_k - \xi^*)]$. The rest of the proof follows
similarly by considering the Lyapunov function $V_{Q_\alpha}$ instead where $Q_\alpha := P_\alpha + 2\alpha\sigma_2^2 C^\top C$. Moreover, we can derive
an extended version of Lemma 3.3, for the case $\sigma_2 > 0$, showing that $\mathbb{E}[V_{Q_{\alpha_{k+1}}}(\xi_1^{k+1})] \leq (2 + 4\alpha\sigma_2^2/\mu)\mathbb{E}[V_{Q_{\alpha_k}}]$.
**Reviewer #2**: **Comments**: We thank the reviewer for positive and insightful comments. We will fix the typo in Eq. (32).
The aim of Corollary 3.2 is to provide an immediate result of Theorem 3.1 and also show the need for a multistage
scheme for achieving the *optimal* bound. We will add more details on this. The reviewer is also absolutely right that
there is a typo in line (132). Since $x_0 = x_{-1}$, as shown in the proof of Lemma 3.3, we can bound the Lyapunov function
by $2(f(x_0) - f^*)$ where the constant 2 is missing. **When $\mu$ is not available**: We thank the reviewer for pointing out
this case. Please see the second part of our response to Reviewer #3. In particular, in Theorem 1 below, we show how
our analysis can directly imply an immediate performance bound for convex objective functions. This result can also be
used when $\mu$ is not available. We will add this result with a complementary discussion to our paper.
**Reviewer #3**: Indeed [8] studies both convex and strongly convex cases. Our focus in this paper is to obtain the optimal
rate for strongly convex functions. In what follows, we first summarize the differences of our work with $\mu$-AGD for
the case of strongly convex objectives and then briefly explain how our results can be directly applied for the convex
case as well. **Comparison with [8]**: As the authors in [8] explain in Corollary B.5 and the discussion after that, their
error bound for strongly convex objective functions, after $n$ iterations, is given by $\mathcal{O}(\frac{p+1}{n^{p+1}}\frac{(L-\mu)\|x_0-x^*\|^2}{2} + \frac{(p+1)^2}{pn}\frac{\sigma^2}{\mu})$
where $p$ is a positive integer. Hence, $\mu$-AGD does not achieve the optimal bias and variance terms simultaneously.
Moreover, given the number of iterations $n$, the authors suggest choosing $p = \log(n)$ which leads to super-polynomial
term in bias (yet not exponential) while the variance term would be a logarithmic factor off from optimal. However, by
Theorem 3.4, our algorithm admits the bound $\mathcal{O}(\frac{(p\sqrt{\kappa})^p exp(-n_1/\sqrt{\kappa})}{n^p}(f(x_0) - f^*) + \frac{p}{n}\frac{\sigma^2}{\mu})$ for any $p \geq 2$. This result
not only recovers the $\mu$-AGD result by choosing $n_1 = p\sqrt{\kappa}\log(\kappa p)$, but also, for a given number of iterations $n$, can
achieve the optimal bias and variance terms simultaneously by choosing $p = 2$ and $n_1 = \mathcal{O}(\frac{n}{C})$ for some constant
$C \geq 2$. **Results for the convex case**: For unconstrained optimization, and without the knowledge of noise parameter
$\sigma^2$, [8] achieves the rate $\mathcal{O}(\frac{1}{\sqrt{n}})$ in both bias and variance terms (see last part of Corollary 3.9 and also Corollary 4.1 in
[8]). As we state below, a direct application of our current results recovers a similar result to [8] up to a log factor. We
leave achieving the optimal rate for convex case for future work.
**Theorem 1.** *Let $f$ be a convex function. Consider running M-ASG for one stage with $n$ iterations and stepsize*
$\alpha_1 = \frac{(\log n)^2}{n^{3/2}L}$. *Then,* $\mathbb{E}\left[f(x_{n+1}^1)\right] - f^* \leq 2/\sqrt{n}(f(x_0) - f^* + L\|x_0 - x^*\|^2) + \sigma^2 \log n/(\sqrt{n}L)$ *for $n \geq 2$.*
*Proof.* We provide a sketch of the proof, and will add more details in our paper. Let $f_\lambda(x) := f(x) + \lambda/2\|x - x_0\|^2$
with $\lambda = L/(\sqrt{n} - 1)$. Note $f_\lambda \in S_{\lambda,L+\lambda}$, and thus, using Theorem 3.1 with $c = \log n/n^{3/4}$ and $\kappa = \sqrt{n}$ implies

$$\mathbb{E}\left[f_\lambda(x_{n+1}^1)\right] - f_\lambda^* \leq \mathbb{E}\left[V_{P_\alpha}(\xi_{n+1})\right] \leq \exp(-n\frac{c}{\sqrt{\kappa}})\mathbb{E}\left[V_{P_\alpha}(\xi_1)\right] + \frac{\sigma^2\sqrt{\kappa}c}{L+\lambda} \leq \frac{1}{n}\mathbb{E}\left[V_{P_\alpha}(\xi_1)\right] + \frac{\sigma^2 \log n}{\sqrt{n}L}.$$

Now, using the fact that $x_0 = x_{-1}$, and similar to the proof of Lemma 3.3, we can show $\mathbb{E}[V_{P_\alpha}(\xi_1)] \leq 2(f_\lambda(x_0)-f_\lambda^*) =$
$2(f(x_0) - f_\lambda^*)$. Using this, along with $f(x_{n+1}^1) \leq f_\lambda(x_{n+1}^1)$, implies $\mathbb{E}[f(x_{n+1}^1)] - (1 - 2/n)f_\lambda^* \leq 2/nf(x_0) +$
$\sigma^2 \log n/(\sqrt{n}L)$. Finally, using the bound $f_\lambda^* \leq f_\lambda(x^*) = f^* + \lambda/2\|x_0 - x^*\|^2$ completes the proof. $\square$
In addition, we can improve this result in term of the dependency to $n$ for the bounded domain case with using a
projection at each step (see Section 5.4 in [23] for a similar result in the deterministic case). The main idea is to use
the argument above in a multistage scheme with decreasing $\lambda$ while going from one stage to the next one. Using the
bounded domain assumption, we can rewrite Lemma 3.3 to stitch stages together.

[Meta-Review · NeurIPS 2019]

This paper designs a multistage SGD algorithm that does not need to know noise and optimality gap at initialization and yet obtain optimal convergence rates. This is a well written paper with good results.